# Self-limiting stem-cell niche signaling through degradation of a stem-cell receptor

**Sophia Ladyzhets**[1]©, **Matthew Antel**[1]©, **Taylor Simao**[1]©, **Nathan Gasek**[1], **Ann E. Cowan**[2,3], **Mayu Inaba**[1]*

**1** Department of Cell Biology, University of Connecticut Health Center, Farmington, Connecticut, United States of America, **2** Richard D. Berlin Center for Cell Analysis and Modeling, University of Connecticut Health Center, Farmington, Connecticut, United States of America, **3** Department of Molecular Biology and Biophysics, University of Connecticut Health Center, Farmington, Connecticut, United States of America

© These authors contributed equally to this work.
* inaba@uchc.edu

**Data Availability Statement:** All relevant data are within the paper and its Supporting Information files.

**Funding:** This work was supported by an NIH grant 1R35GM128678-01 and start-up funds from

## Abstract

Stem-cell niche signaling is short-range in nature, such that only stem cells but not their differentiating progeny receive self-renewing signals. At the apical tip of the *Drosophila* testis, 8 to 10 germline stem cells (GSCs) surround the hub, a cluster of somatic cells that organize the stem-cell niche. We have previously shown that GSCs form microtubule-based nanotubes (MT-nanotubes) that project into the hub cells, serving as the platform for niche signal reception; this spatial arrangement ensures the reception of the niche signal specifically by stem cells but not by differentiating cells. The receptor Thickveins (Tkv) is expressed by GSCs and localizes to the surface of MT-nanotubes, where it receives the hub-derived ligand Decapentaplegic (Dpp). The fate of Tkv receptor after engaging in signaling on the MT-nanotubes has been unclear. Here we demonstrate that the Tkv receptor is internalized into hub cells from the MT-nanotube surface and subsequently degraded in the hub cell lysosomes. Perturbation of MT-nanotube formation and Tkv internalization from MT-nanotubes into hub cells both resulted in an overabundance of Tkv protein in GSCs and hyperactivation of a downstream signal, suggesting that the MT-nanotubes also serve a second purpose to dampen the niche signaling. Together, our results demonstrate that MT-nanotubes play dual roles to ensure the short-range nature of niche signaling by (1) providing an exclusive interface for the niche ligand-receptor interaction; and (2) limiting the amount of stem cell receptors available for niche signal reception.

## Introduction

Many stem cells reside in a special microenvironment, called the niche, to maintain their identity [1]. In the *Drosophila* testis, germline stem cells (GSCs) reside in a niche formed by postmitotic somatic cells called hub cells. GSCs typically divide asymmetrically, giving rise to 1 daughter cell that retains its attachment to the hub and self-renews, while the other daughter cell, a gonialblast (GB), is displaced away from the hub and differentiates into spermatogonia (SG). Hub cells secrete the ligands Decapentaplegic (Dpp) and Unpaired (Upd). Dpp activates

UConn Health (to M.I.). The funders had no role in study design, data collection and analysis, decision to publish, or preparation of the manuscript.

**Competing interests:** The authors declare no competing interests.

**Abbreviations:** Bam, Bag of marbles; BMP, Bone Morphogenetic Protein; BSA, bovine serum albumin; CC, cyst cell; CQ, chloroquine; CySC, somatic cyst stem cell; DAPI, 4,6-diamidino-2-phenylindole; Dpp, Decapentaplegic; DSHB, Developmental Studies Hybridoma Bank; ESCRT, endosomal sorting complexes required for transport; FRAP, fluorescence recovery after photobleaching; GB, gonialblast; Gbb, Glass bottom boat; GSC, germline stem cell; FISH, fluorescence in situ hybridization; HECT, homologous to the E6-AP carboxyl terminus; HSPG, heparan sulfate proteoglycan; IFT-B, intraflagellar transport-B; JAK-STAT, Janus kinase-signal transducer and activator of transcription; lamp1, Lysosomal-associated membrane protein-1; MT-nanotubes, microtubule-based nanotubes; PBS, phosphate-buffered saline; pMad, phosphorylated Mad; RNAi, RNA interference; RPE, retinal pigment epithelium; RT-PCR, reverse transcription PCR; SG, spermatogonia; spin, spinster; Smurf, SMAD ubiquitination regulatory factor; Tkv, Thickveins; Upd, Unpaired; VDRC, Vienna Drosophila Resource Center.

the Bone Morphogenetic Protein (BMP) pathway, whereas Upd activates the Janus kinase-signal transducer and activator of transcription (JAK-STAT) pathway in GSCs, both of which are required for maintenance of GSCs [2–7]. These niche-derived ligands must act over a short range so that signaling is only active in GSCs but not in GBs. Defining the boundary of niche signaling between abutting GSCs and GBs is of critical importance in maintaining stem cell populations while ensuring differentiation of their progeny. Nevertheless, how the short-range nature of niche signaling is achieved is poorly understood.

We have previously demonstrated that cellular projections called microtubule-based (MT)-nanotubes present specifically on GSCs and project into hub cells (Fig 1A) [8]. The receptor for the Dpp ligand, Thickveins (Tkv), is produced by GSCs and trafficked to the surface of MT-nanotubes where it interacts with hub-derived Dpp, which led us to propose that MT-nanotubes serve as a signaling platform for productive Dpp–Tkv interaction (Fig 1A). Here we show that MT-nanotubes serve a second function: In addition to serving as a platform for Dpp–Tkv interaction, MT-nanotubes also promote degradation of Tkv within hub cells. We observed Tkv expressed in GSCs localizes to lysosomes in hub cells. Perturbation of MT-nanotube formation compromises Tkv localization in hub lysosomes, resulting in an overabundance of Tkv within GSCs. Moreover, germline-specific expression of a mutant form of Tkv that cannot localize in hub-cell lysosomes enhanced downstream signal activation in GSCs. These data suggest that Tkv degradation in the hub plays a role in attenuating the niche signal. We propose here that MT-nanotubes not only promote niche signal reception by stem cells, but also modulate the signal strength to an appropriate level via degradation of the receptor.

## Results

### Tkv expressed in GSCs is transferred to hub lysosomes

We have previously shown that Tkv is produced in GSCs and trafficked to the surface of MT-nanotubes. If MT-nanotubes are reduced or eliminated, Tkv remains with the GSC cell body plasma membrane and Dpp–Tkv interactions are significantly reduced, as demonstrated by a reduction in downstream activation [8].

In addition to Tkv localization on MT-nanotubes, we noticed that a majority of the Tkv-mCherry signal was observed within the cell body of hub cells even though it was expressed specifically in the germline (*nosGal4>tkv-mCherry*) (Fig 1B, S1 Movie). Moreover, we found that Tkv-mCherry within hub cells did not always colocalize with MT-nanotubes (Fig 1C, white arrowheads), indicating that Tkv on the MT-nanotubes might be internalized into hub cells. To test if the Tkv seen in hub cells originated from GSCs, we examined the time course of Tkv localization. When expression of Tkv-mCherry was initiated by heat shock (*hs-flp*, *nos-FRT-stop-FRT-Gal4*, *UAS-GFP*, *UAS-tkv-mCherry*), Tkv-mCherry was first observed on the GSC plasma membrane and in the GSC cytoplasm as puncta (day 1 after heat shock). After 2 days post heat shock, its signal in the hub became evident. Finally, after 3 days, the Tkv signal along the GSC plasma membrane was reduced and the Tkv signal in the hub increased further (Fig 1D–1F). These results indicate that Tkv seen in hub area are derived from GSCs.

We found that Tkv punctae observed in hub cells often colocalize with lysosomes (Fig 1G and 1H). Localization of Tkv in lysosomes was further examined by using chloroquine (CQ) treatment, a drug that inhibits lysosomal enzymes and increases the size of lysosomes (Fig 1I–1K) [9–12]. When testes were treated with CQ, we observed that both endogenously tagged Tkv (Tkv-GFPtrap) and germline expressed Tkv-GFP (*nosGal4>tkv-mCherry*) localized to enlarged punctae in the hub (Fig 1L–1Q), confirming that a large proportion of Tkv-positive punctae are lysosomes.

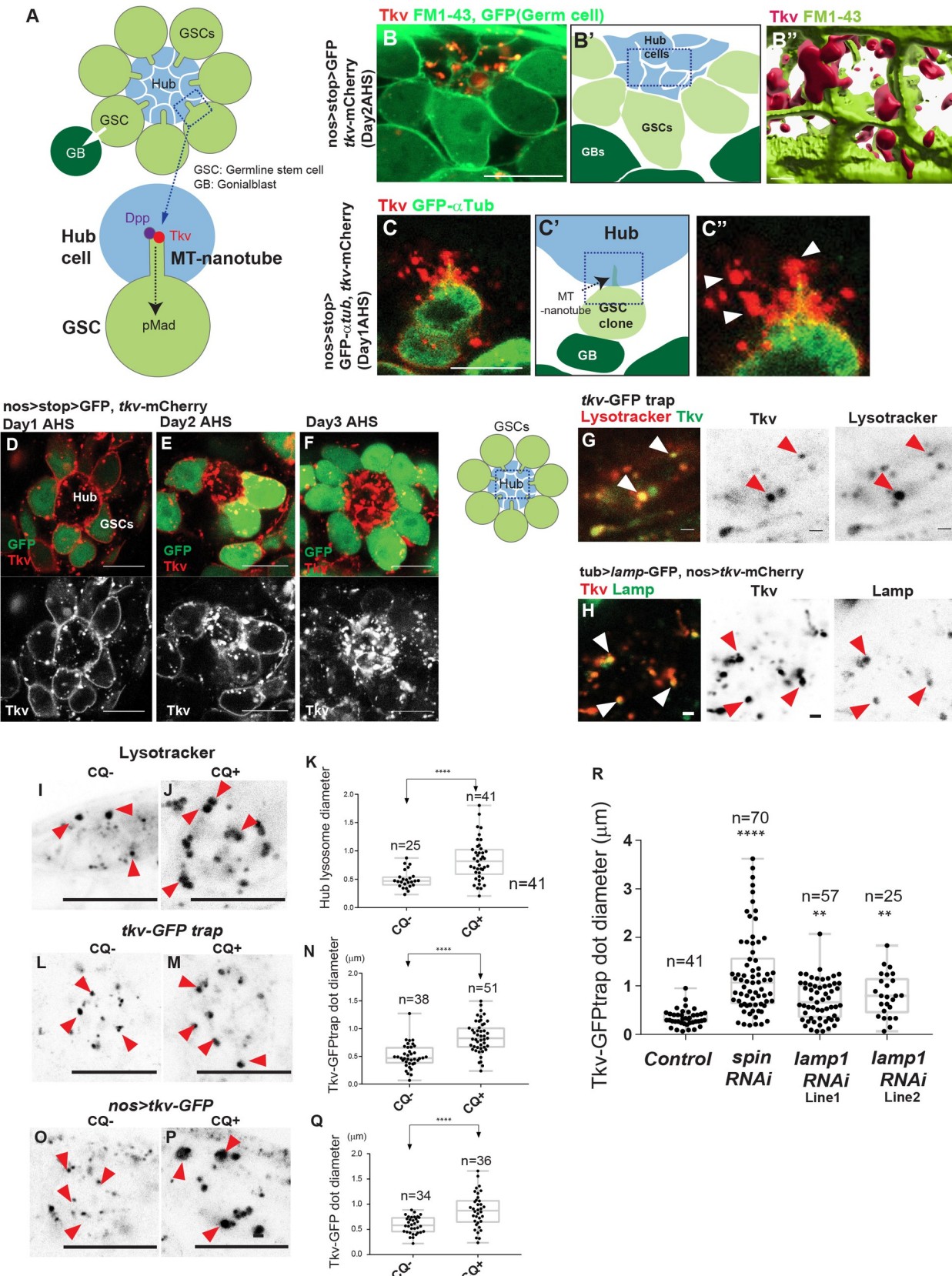

**Fig 1. Tkv expressed in GSCs is transferred to hub lysosomes. A, Top,** Schematic of the *Drosophila* male GSC niche. GSCs are attached to the hub. The immediate daughters of GSCs, the GBs, are displaced away from the hub then start differentiation (a white arrow indicates the division orientation). **Bottom,** Schematic of Dpp ligand-Tkv receptor interaction on the surface of an MT-nanotube, leading to pMad. **B,** A representative image of a hub and surrounding GSCs expressing Tkv-mCherry (red) and GFP (green). Plasma membranes of cells are visualized by FM1-43 dye (green). Tkv expressed in GSCs (red) is seen in the hub cells. A testis from *hs-flp, nos-FRT-stop-FRT-Gal4, UAS-tkv-mCherry, UAS-GFP* fly was imaged 2 days after heat shock. **B'** A graphic interpretation of the image **B**. Blue box indicates the area used for three-dimensional rendering in **B''**; **B''** A three-dimensional rendering of the hub area (blue box in **B'**). **C,** A GSC clone expressing Tkv-mCherry (red) and GFP-αTub (green). Testes from *hs-cre, nos-loxP-stop-loxP-Gal4, UAS-tkv-mCherry, UAS-GFP-αtub* flies were used 1 day after heat shock. **C',** A graphic interpretation of the image **C**. The blue box shows the area corresponds to the magnified region shown in **C''**. **C'',** A magnified view of the blue box in **C''**. White arrowheads in **C''** indicate Tkv-mCherry puncta that are distant from the MT-nanotube. **D–F,** Representative images of the hub and surrounding GSCs (expressing GFP, green) at the indicated days after induction of Tkv-mCherry (red) expression. Testes from *hs-flp, nos-FRT-stop-FRT-Gal4, UAS-tkv-mCherry, UAS-GFP* flies were imaged indicated days AHS. **G, H,** Representative images of the hub area of indicated genotypes. Arrowheads indicate the colocalization of Tkv with hub lysosomes (marked by lysotracker in **G**, Lamp-GFP in **H**). **I, L,** Representative images of lysotracker-positive lysosomes in the hub in 4-hour cultured testes without (**I**, CQ−) or with (**L**, CQ+) chloroquine. Red arrowheads indicate hub lysosomes. **K,** Measured diameters of lysosomes in the hub with or without CQ. **L, M,** Representative images of the hub area of tkv-GFPtrap testes in 4-hour cultured testes without (**L**, CQ−) or with (**M**, CQ+) chloroquine. Red arrowheads indicate Tkv-positive punctae. **N,** Measured diameters of Tkv punctae in the hub with or without CQ. **O, P,** Representative images of the hub area surrounded by Tkv-GFP-expressing GSCs under the nosGal4 driver in 4-hour cultured testes without (**O**, CQ−) or with (**P**, CQ+) chloroquine. Red arrowheads indicate Tkv-positive punctae. **Q,** Measured diameters of Tkv punctae in the hub with or without CQ. **R,** Diameters of Tkv-GFPtrap punctae in the hub of indicated genotypes (the largest diameter for each punctum was chosen from a z-stack collected at 0.5 µm intervals). Images **G, H, I, L, Q,** show a portion of the hub (depicted as a blue box in the left-hand panel of **G**). In **K, N, Q, R,** the largest diameter of each puncta was chosen from z-stacks collected at 0.5 µm intervals from the hub area of each genotype. Indicated numbers of punctae from 2 independent experiments were scored for each group. *P* values (adjusted *P* values from Dunnett multiple comparisons test) are provided as \*\**P* < 0.01, \*\*\*\**P* < 0.0001. The box plot shows 25%–75% (the box), the median (horizontal band inside the box), and minima to maxima (whiskers). All imaging and measurements were performed using live tissues. Underlying numerical data for **K, N, Q,** and **R** are provided in S1 Data. CQ, chloroquine; days AHS; days after heat shock; Dpp, Decapentaplegic; GB, gonialblast; GFP, green fluorescent protein; GSC, germline stem cell; MT-nanotube, microtubule-based nanotube; pMad, phosphorylation of a downstream effector, Mad; Tkv, Thickveins.

In addition to Tkv, the type II receptor Punt, a co-receptor of Tkv, was observed in hub lysosomes (S1A–S1C Fig). Moreover, Dpp ligand (visualized via a knock-in line in which endogenous Dpp is fused to mCherry [13]) colocalized with Tkv-GFPtrap in the hub (S1D Fig) and was also seen in lysosomes in the hub lysosomes (S1B Fig) as was a reporter of ligand-bound Tkv, TIPF [14] (*nosGal4>TIPF*) (S1C Fig).

It should be noted that the endogenously tagged Tkv (Tkv-GFPtrap) signal exhibited complete overlap with the Tkv-mCherry transgene expressed in the germline (*nosGal4>tkv-mCherry*) (S1E Fig), indicating that the Tkv-GFPtrap signal seen in the hub entirely originated in GSCs, and Tkv-mCherry localization in hub lysosomes is unlikely due to an overexpression artifact.

To determine if Tkv containing lysosomes are made in hub cells, we knocked down *spinster* (*spin*) and *Lysosomal-associated membrane protein-1* (*lamp1*) genes using a hub cell-specific driver (*updGal4*). Spin is a putative late-endosomal/lysosomal efflux permease [15]. Lamp1 is an abundant protein in the lysosomal membrane and is required for lysosomes to fuse with endosomes [16]. *spin* mutations have been shown to enlarge the size of lysosomes [17]. Consistently, we observed that hub cell-specific knockdown of *spin* (*updGal4>spin RNAi*) enlarged the size of Tkv-positive puncta in hub cells (Fig 1R), suggesting that the Tkv containing lysosomes seen in the hub area are generated in hub cells. Similarly, *lamp-1* knockdown also resulted in enlarged sizes of Tkv-positive lysosomes (Fig 1R).

Together, our data suggest that the Tkv receptor expressed in GSCs localizes to MT-nanotubes then is translocated into lysosomes located in hub cells with other signaling components.

## MT-nanotubes are required for Tkv transfer from GSCs to hub cells

*Intraflagellar transport-B* (*IFT-B*; *oseg2*, *osm6*, and *che-13*) gene products are required for MT-nanotube formation (Fig 2A) [8]. As shown in our previous report, disruption of MT-nanotubes by knocking down *IFT-B* (IFT-KD) causes Tkv retention within GSCs (Fig 2B and 2C,

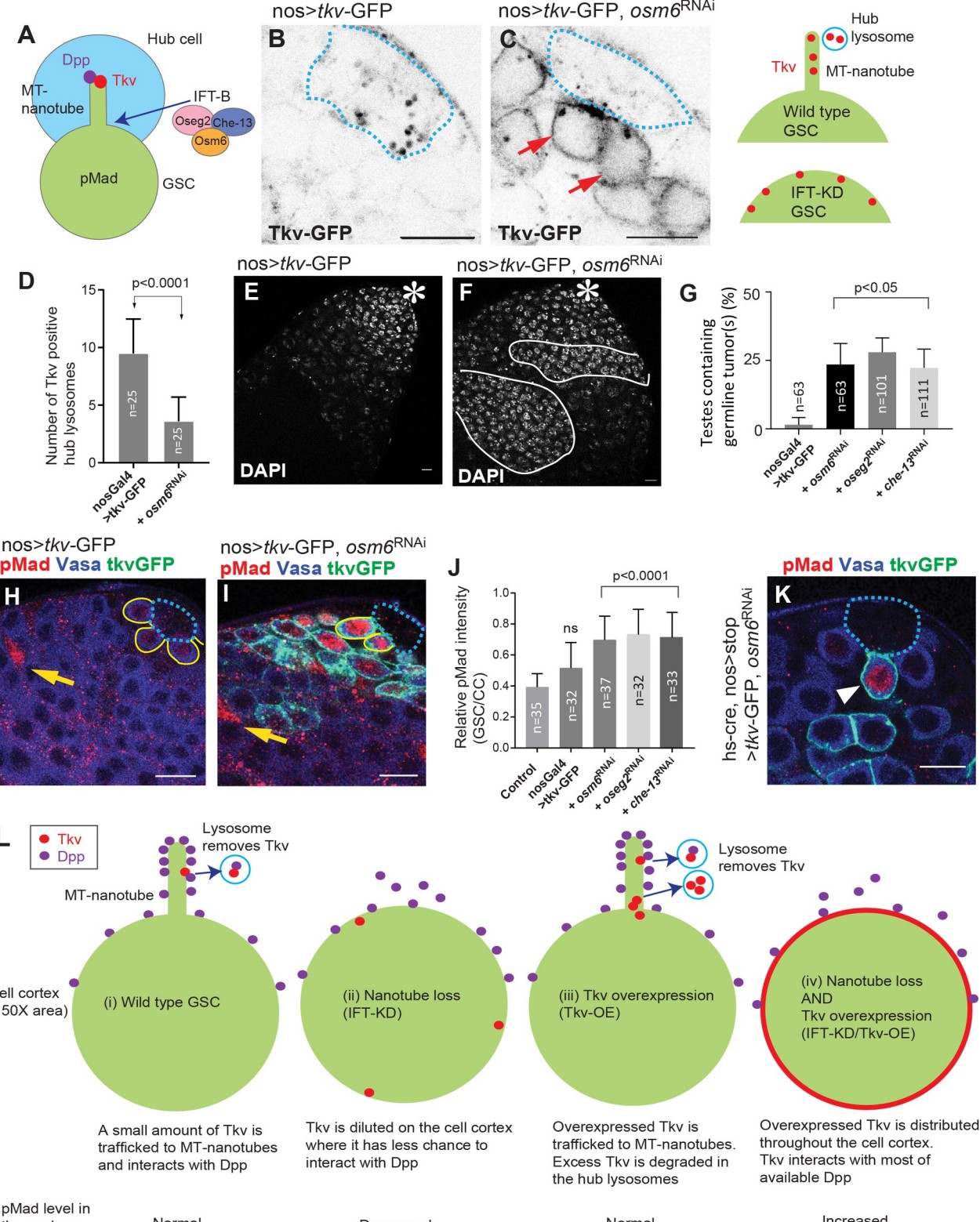

**Fig 2. MT-nanotubes are required for Tkv transfer from GSCs to hub cells. A,** A schematic of MT-nanotube formation. IFT-B proteins (Oseg2, Che-13, and Osm6) are required for MT-nanotube formation. **B,** Representative image of a testis tip of a Tkv-OE fly (*nosGal4>tkv-GFP*). **C,** Representative image of a testis tip of a Tkv-OE/IFT-KD fly (*nosGal4>tkv-GFP, osm6 RNAi*). Red arrows indicate Tkv-GFP localization along the whole cell cortex of GSCs. The right panel explains the Tkv localization pattern in GSCs with or without MT-nanotube formation. In **B** and **C,** blue

dotted lines outline the hub. **D,** Number of Tkv-positive hub lysosomes in the 2 indicated genotypes. Lysosomes in the entire hub region were counted as lysotracker positive punctae >0.5 μm diameter in z-stacks collected at 0.5 μm interval. The *P* value was calculated by Student *t* test. Total testes (*n* = 25) from 2 independent experiments were scored for each group. **E, F,** Representative DAPI staining images of testes of Tkv-OE (*nosGal4>tkv-GFP*) without (**E**) or with (**F**) IFT-KD (*nosGal4>osm6 RNAi*). Asterisks indicate the approximate location of the hub. White lines in **F** outline germline tumors with condensed chromatin detected by DAPI staining (white), separated from the group of cells containing GSCs, GBs, SGs, and somatic cyst stem cells that are typically observed only near the tip of the testis as a single group (**E**). (**G**) Percentage of testes containing one or more germline tumors in the indicated genotypes. The indicated numbers of testes (n) from 3 independent experiments were scored for each group. **H, I,** Representative images of pMad staining in the testes of Tkv-OE (*nosGal4>tkv-GFP*) flies, without (**H**) or with (**I**) IFT-KD (*nosGal4>osm6 RNAi*). Vasa (blue), pMad (red), Tkv (green; note that Tkv overlapping with Vasa appears as cyan). Blue dotted lines outline the hub. Yellow lines outline GSCs. Arrows indicate CCs used as an internal control (see Methods). **J,** Quantification of pMad intensity in the GSCs relative to somatic cyst cells (CCs; yellow arrows in **H, I**). Indicated numbers (n) of GSCs from 3 independent experiments were scored for each group. **G and J,** Adjusted *P* values from Dunnett multiple comparisons test are provided. **K,** A representative image of a GSC clone (arrowhead) expressing Tkv together with IFT-KD. Blue dotted lines outline the hub. Vasa (blue), pMad (red), Tkv (green). **L,** A model explaining how IFT-KD can cause opposite effects on pMad levels in the GSC nucleus depending on Tkv expression level. Scale bars are 10 μm in all images. For **B–D,** imaging and measurements were performed using live tissues. Fixed samples were used for other experiments. Underlying numerical data for **D, G,** and **J** are provided in S1 Data. CC, cyst cell; GB, gonialblast; GSC, germline stem cell; IFT-B, intraflagellar transport-B; IFT-KD, intraflagellar transport-knockdown; MT-nanotubes, microtubule-based nanotubes; SG, spermatogonia; Tkv, Thickveins; Tkv-OE, Tkv overexpression.

and see [8]). Therefore, we hypothesized that MT-nanotubes might be required for Tkv transfer from GSCs to hub cells in addition to their function in Dpp ligand reception. We found that inhibiting MT-nanotube formation by knocking down *IFT-B* genes significantly reduced the number of Tkv-positive hub lysosomes (Fig 2D). The finding that inhibition of MT-nanotube formation resulted in both a reduction of Tkv-positive hub lysosomes as well as the retention of Tkv in the GSC's cell body suggested that MT-nanotubes may serve as the platform not only for Tkv/Dpp interaction, but also for Tkv internalization from GSCs to hub cells. It should be noted that we did not detect increased Tkv in the GSC surface upon IFT knockdown when we used endogenously tagged Tkv (Tkv-GFPtrap), likely due to the low amount of endogenous protein compared to neighboring somatic cell populations (S2A Fig).

Although shutting down the Dpp signal is required for proper differentiation, neither down-regulating Tkv trafficking to the hub (*nosGal4>IFT-B RNAi*, referred to as IFT-KD) nor Tkv overexpression (*nosGal4>tkv*, referred to as Tkv-OE), both of which are expected to increase the available Tkv in GSCs, were alone found to impact differentiation. However, when Tkv-OE was combined with IFT-KD, we often observed ectopic proliferation of undifferentiated germ cells outside of the niche (hereafter referred to as a germline tumor, Fig 2E–2G). Bag of marbles (Bam) is a master differentiation factor whose expression is suppressed by Dpp signaling and typically peaks around the 4- to 8-cell SG stage [18]. In the IFT-KD/Tkv-OE testes, the Bam peak was never observed (S2C and S2D Fig), suggesting that the germline tumor was caused by a failure to shut down Dpp signaling. Moreover, under these conditions, cytoplasmic STAT92E (but not nuclear STAT92E) remained high in both germline and germline tumor cells (S2E and S2F Fig). We observed a similar high STAT92E expression level when a constitutively active form of Tkv was expressed. Therefore, we consider the high cytoplasmic STAT92E level in germ cells after exit from the niche to reflect prolonged Dpp signal activation (S2G Fig).

Because we observed germline tumors far away from the niche, we wondered if the effect of IFT on Tkv down-regulation is specific in GSCs where MT-nanotubes are present. We expressed Tkv-GFP using a bamGal4 driver that is active in 4- to 8-cell spermatogonia with or without IFT-KD (bam>Tkv-OE, or IFT-KD/Tkv-OE). As expected, we did not observe any difference in Tkv-GFP expression level between these 2 conditions (S2H and S2I Fig). Moreover, we never observed germline tumor formation in either genotype (bam>Tkv-OE, *n* = 67, bam>IFT-KD/Tkv-OE, *n* = 114). Together, these data indicate that IFT-KD influences Tkv protein degradation specifically in GSCs.

## MT-nanotube loss enhances Dpp signaling in the presence of Tkv overexpression

Previously, we showed that IFT-KD interferes with Dpp signaling resulting in reduced phosphorylated Mad (pMad, the read-out of Dpp signal activation) in GSCs [8]. However, in this study, we observed that GSCs in IFT-KD/Tkv-OE testes exhibited increased pMad levels compared with GSCs only overexpressing Tkv (Fig 2H–2J). Moreover, the effect of IFT-KD/Tkv-OE to increase pMad is cell-autonomous; GSC clones with Tkv-OE/IFT-KD (*hs-cre*, *nos-loxP-stop-loxP-Gal4> tkv*, *IFT-B RNAi*) but not other neighboring GSCs showed increased pMad levels (Fig 2K). These results indicate that the IFT-KD has an opposite effect on signaling outcome in the presence of Tkv overexpression.

The opposite outcomes of MT-nanotube perturbation on Dpp signal activation (seen as nuclear pMad levels in GSCs) can be explained by different Tkv expression levels. Under normal conditions, Tkv is trafficked to the surface of MT-nanotubes, where it interacts with Dpp, leading to a "normal" pMad level in the GSC nucleus (Fig 2L-i). When MT-nanotube formation is compromised, Tkv disperses to the entire cell cortex, which might reduce its effective interaction with Dpp leading to reduced pMad levels. If the same number of molecules A and B are located on the areas of different sizes, the binding probability between A and B is proportional to approximately 1/area size. Based on our confocal images, the nanotube comprises about 1/150 of the surface of the entire cell. If the Tkv destined for the MT-nanotube surface redistributes to the entire cell plasma membrane, we would predict a 1/150 decrease in binding rate of Tkv and Dpp (Fig 2L-ii). This also suggests the possibility that MT-nanotubes may contribute to a higher probability of ligand–receptor interaction by concentrating molecules on their surface.

When Tkv is overexpressed, excess Tkv is trafficked to MT-nanotubes and degraded, and no increased pMad is observed (Fig 2L-iii). When Tkv is overexpressed and the MT-nanotube is compromised, excess Tkv localizes to the entire cell cortex and is no longer degraded. Tkv on the cell cortex now interacts with more of the available Dpp leading to an increased pMad level (Fig 2L-iv).

## Inhibition of hub lysosomes results in enhancement of Dpp-Tkv signaling

Next, we investigated the signaling outcome of impaired lysosome function in hub cells. It has been reported that Dpp-Tkv signaling is enhanced in *spin* mutant eye discs where Spin functions on lysosomes that likely target signaling endosomes for degradation [19]. Hub cell-specific knockdown of lysosomal genes (*updGal4>spin RNAi or lamp1 RNAi*) led to a significant increase in the levels of pMad in GSCs (Fig 3A, 3B and 3E), indicating that hub lysosomes function in Dpp signal attenuation. In contrast, germline-specific knockdown (*nosGal4>spin RNAi or lamp1 RNAi*) did not alter pMad levels in GSCs (Fig 3C–3E). Note that pMad staining in somatic cyst cells (CCs) was unaffected by inactivation of hub lysosomes (S3A Fig) and could therefore be used to normalize pMad staining intensity.

In addition to enlarged Tkv-positive lysosomes mentioned above (Fig 1R), we noticed that the Tkv-GFPtrap signal often showed a thread-like pattern in the hub cells expressing Spin or Lamp1 RNA interference (RNAi) (Fig 3F–3H). Thread-like Tkv was typically not associated with lysosomes (Fig 3I), and the frequency of emergence of a Tkv thread was significantly higher than control (Fig 3J). We also observed a similar thread-like Tkv pattern after CQ treatment (Fig 3K). When Tkv-mCherry was expressed together with αTub-GFP to visualize MT-nanotubes, thread-like Tkv was often observed along with αTub-GFP-positive MT-nanotubes (Fig 3L). The frequency of hubs containing Tkv-decorated MT-nanotubes was significantly increased after CQ treatment (Fig 3M). These data indicate that the thread-like Tkv pattern observed after inhibition of hub likely represents the increased Tkv on MT-nanotubes.

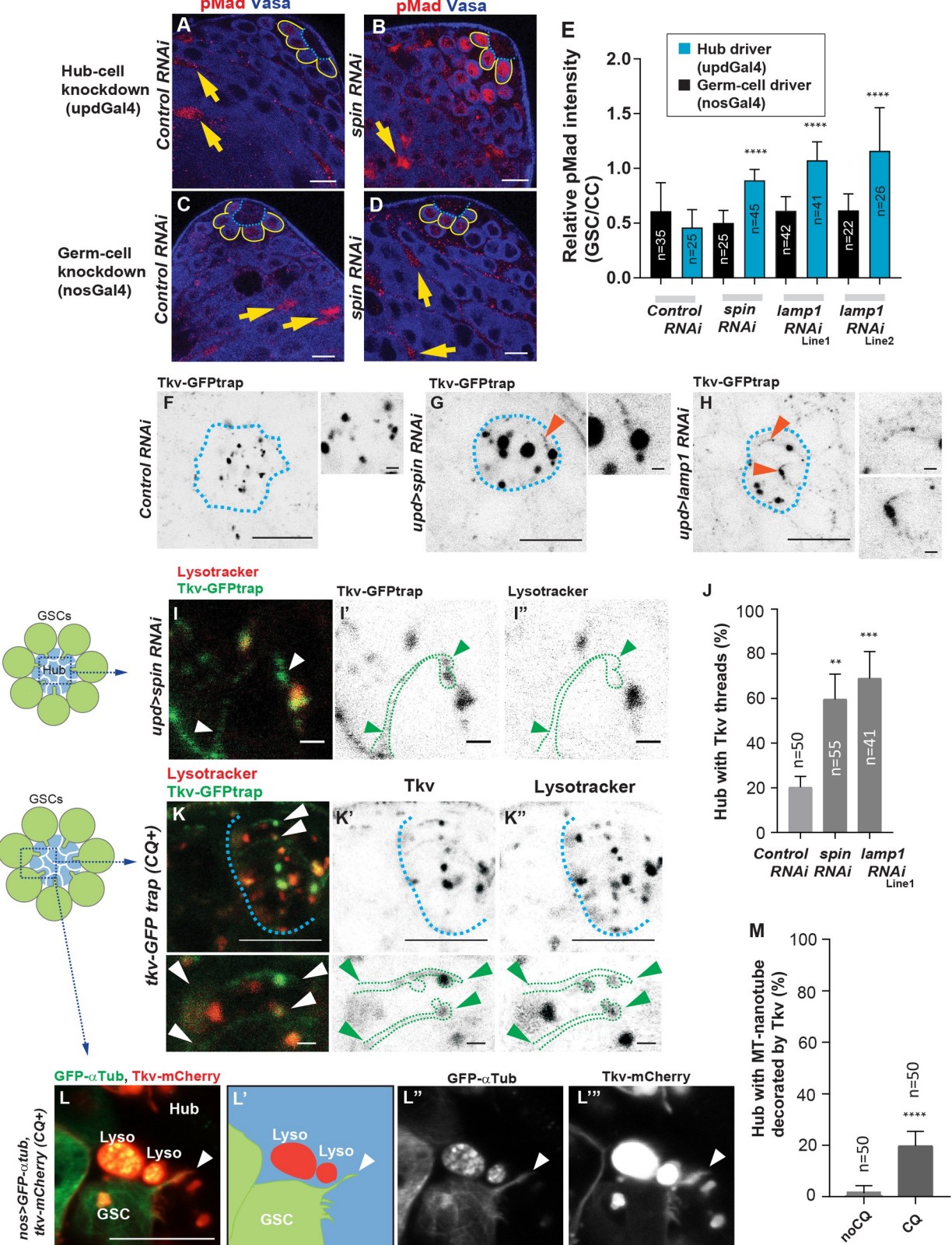

**Fig 3. Inhibition of hub lysosomes results in enhanced Dpp-Tkv signaling. A–D,** Representative images of pMad staining in the testis of indicated genotypes. A driver specific for hub cells (*updGal4*, **A, B**) or germ cells (*nosGal4*, **C, D**) was used to induce RNAi. Blue dotted lines

outline the hub. Yellow lines outline GSCs. pMad (red). Vasa (blue, germ cell marker). The mRNA level was reduced in each RNAi line as follows: Lamp1 Line1; TRiP.HMS01802 47.79%, Line2; TRiP GLV21040 50.87%, Spin 25.05%, see Methods. **E,** Quantification of pMad intensity in the GSCs relative to somatic CCs (yellow arrows in **A–D**). Note that pMad in CCs remained the same in indicated genotypes (see S2A Fig). Numbers (n) of GSCs from 3 independent experiments were scored for each group. **F–H,** Examples of the thread pattern of the tkv-GFPtrap signal (Tkv threads) seen in the hub of indicated genotypes. Blue dotted lines outline the hub. Arrowheads indicate Tkv threads seen after the knockdown of hub lysosomal genes (*spin*, *lamp1*). Magnified portions of the hub are shown in right-hand panels. **I,** Representative images of a Tkv thread in a portion of the hub of a *spin* RNAi fly. Tkv threads are encircled by green dotted lines in **I', I"**. Arrowheads indicate the starting and ending points of the thread. Tkv threads are typically lysotracker negative. Tkv-GFPtrap (green), Lysotracker (red). The image in **I** is a portion of the hub marked as a blue box in the left-hand panel. **J,** Frequency of hubs containing any Tkv threads in each genotype. **K,** Representative image of Tkv threads and associating lysotracker-positive lysosomes in a portion of the hub of a CQ-treated testis (4 hours). Tkv threads are typically lysotracker negative. Tkv-GFPtrap (green), Lysotracker (red). Blue dotted lines outline the hub. Lower panels are magnified images of the threads. Tkv threads are encircled by green dotted lines. Arrowheads indicate end points of the threads. Tkv-GFPtrap (green), Lysotracker (red). **L,** Representative image of Tkv threads on an MT-nanotube in a testis expressing GFP-αTub together with Tkv-mCherry (*nosGal4>GFP-αtub*, *tkv-mCherry*). MT-nanotube (GFP-αTub, green), Tkv-mCherry (red). A white arrowhead points the tip of a Tkv-decorated MT-nanotube. Images in **K** and **L** are the portion of the hub marked as a blue box in the left-hand panel of **K**. **M,** Frequency of hubs containing any MT-nanotubes decorated by Tkv in each genotype. In **J** and **M**, indicated numbers (n) of testes from 2 independent experiments were scored for each group. Entire hub areas were imaged for scoring using z-stacks collected at 0.5 μm intervals. Scale bars; 1 μm in right-hand panels in **F–H**, **I**, and **K**. 10 μm in other images. In **A–E**, fixed samples were imaged; in **F–M**, imaging and measurements were performed using live tissues. For **E**, **J**, and **M**, P values (adjusted P values from Dunnett multiple comparisons test) are provided as **P < 0.01, ***P < 0.001, ****P < 0.0001, non-significant (p≥0.05) if not shown. Underlying numerical data for **E**, **J**, and **M** are provided in S1 Data. CC, cyst cell; CQ, chloroquine; Dpp, Decapentaplegic; GSC, germline stem cell; *lamp1*, Lysosomal-associated membrane protein-1; MT-nanotubes, microtubule-based nanotubes; pMad, phosphorylated Mad; RNAi, RNA interference; *spin*, spinster; Tkv, Thickveins.

It is still unclear how a lysosomal defect causes increased amounts of Tkv on MT-nanotubes. Similar to this case, it has been reported that a degradation defect of endocytosed Tkv subsequently increases the Tkv protein on the plasma membrane [20], suggesting that there is a potential feedback mechanism such that a lysosomal defect inhibits further endocytosis in the cells, leading to increased Tkv on the cell membrane and signal hyperactivation.

Together, these data, the increase of both pMad in GSCs and Tkv on MT-nanotubes upon inactivation of hub lysosomes, suggest that hub lysosomes may function to down-regulate Dpp signaling by modulating Tkv internalization from MT-nanotubes.

Because interference with lysosomal function may have broad effects, we next tested the possibility that other factors might indirectly influence pMad levels in GSCs. Upd, another major hub ligand, activates the JAK-STAT pathway in GSCs and surrounding somatic cyst stem cells (CySCs). The amount of nuclear STAT92E in GSCs and CySCs did not show a detectable change after inactivation of hub lysosomes (S3B–S3D Fig), suggesting that hub lysosomal activity does not have an impact on either the interaction of Upd ligand/receptor or the activation of the JAK-STAT pathway. It should be noted that the cytoplasmic STAT92E level in differentiating germ cells was higher after hub-specific knockdown of lysosomal genes than that of control testes (S3E and S3F Fig) and similar to testes expressing Tkv-OE/IFT-KD (S2F Fig) or constitutively active Tkv (*nosGal4>tkv-CA*, S2G Fig), suggesting that enhanced Dpp signal activation results in high STAT92E expression.

It has been reported that dpp mRNA is present in CySC populations [7], and JAK-STAT signaling from hub cells induces an increase in dpp mRNA in CySCs [5,21]. We tested whether CySC numbers were increased and/or whether CySCs might express a higher amount of dpp mRNA upon lysosome inactivation, which could increase pMad levels in GSCs. We found that the number of CySCs remained the same upon hub specific knockdown of lysosomal genes (S3G–S3J Fig). A fluorescence in situ hybridization (FISH) experiment using a fluorescent probe for dpp mRNA showed no change in dpp mRNA levels in the hub upon inactivation of hub lysosomes (S4 Fig). Furthermore, we could not detect dpp mRNA in CySCs in either controls or after hub specific knockdown of lysosomal genes (S4 Fig), indicating that dpp transcripts are also unlikely to be increased in CySCs. Together, these results suggest that the increased Dpp signal observed after inhibition of hub lysosomes is not caused by increased

dpp mRNA levels. Note that the dpp mRNA was also reported to be undetectable in wild-type CySCs using in situ method [21]. We speculate that the level of dpp transcript in CySCs must be low such that it is only detectable using reverse transcription (RT)-PCR, as described in the original report [7].

## Inhibition of hub lysosomes does not impact Dpp diffusion

Since we detected the Dpp ligand in hub lysosomes (S1B Fig), we next tested whether the level of Dpp protein level changes after hub-specific lysosomal inhibition. It should be noted that another BMP ligand, Glass bottom boat (Gbb), has also been reported to bind to Tkv and cooperatively maintain GSCs. Gbb is broadly expressed in somatic cells outside of the niche, whereas Dpp is concentrated in the hub [7]. Unlike ectopic Dpp expression (*nosGal4>dpp*), which is known to cause germline tumors, Gbb expression (*nosGal4>gbb*) does not cause any phenotype in the testis [4]. Therefore, we focused on Dpp as a potential rate-limiting factor for the niche-derived BMP signaling in GSCs.

First, to investigate the range of Dpp penetration from the hub, we examined the distribution of Dpp-mCherry (mCherry knock-in strain [13]) together with Tkv-GFPtrap after lysosomal inhibition. Intriguingly, after CQ treatment, we broadly observed Dpp-mCherry containing cells located in the entire anterior region of the testis (Fig 4A and 4B). The majority of detected Dpp-mCherry–positive punctae in these cells were thought to be lysosomes as they became visible only after CQ treatment. These lysosomes were also positive for Tkv-GFPtrap, suggesting that Dpp bound to Tkv on cells located away from the hub are fated to be degraded in their lysosomes (Fig 4A and 4B). Consistently, TIPF, a reporter of ligand-bound Tkv, expressed in the germline (*nosGal4>TIPF*), was also observed in differentiating germ cells (Fig 4C). Dpp-positive lysosomes were observed mainly in CySCs (Fig 4D), but also in 2-cell stage SGs at a lower intensity (Fig 4D). Importantly, we barely observed Dpp/Tkv-positive lysosomes within GSCs, further suggesting that GSCs may not digest Tkv in their own lysosomes (Fig 4D).

Next, to test if Dpp can diffuse from the hub, we expressed Dpp-mCherry specifically in the hub (*updGal4^ts>dpp-mCherry*, 4-day temperature shift). We observed a strong signal in the hub. In addition, a highly dynamic mCherry signal outside of the hub was also detected, likely reflecting Dpp-mCherry diffusing from the hub. To determine if the fluorescence detected outside the hub was consistent with diffusion of Dpp-mCherry in the extracellular space, we used fluorescence recovery after photobleaching (FRAP) analysis. After photobleaching a circle of 5 μm diameter located approximately 10 μm away from the hub, the photobleached region recovered and reached to the plateau at the time 40 seconds after photobleaching. The bleached signal was not recovered to the original level. The maximum intensity value was 53.9% ± 3.18% of original intensity in average (*n* = 3) (Fig 4F and 4H, S2 Movie). The high amount of background signal in the testis (see Fig 4J, S5 Fig for an example of fluorescent intensity using a control testis not expressing Dpp-mCherry) likely accounts for the immobile fraction in this experiment. Recovery times of seconds in these experiments is most consistent with transport of the Dpp-mCherry signal outside of the hub by diffusion in the extracellular space.

In contrast, after photobleaching the strong signal seen in the entire hub region, most of the signal did not recover over the time scale of the experiment (Fig 4G and 4I, S3 Movie), indicating that the majority of Dpp-mCherry seen in the hub is localized within the cells (either newly formed proteins or the protein already in lysosomes).

Finally, we examined if Dpp protein level is altered upon inhibition of hub lysosomes. When Spin was knocked down in the hub cells expressing Dpp-mCherry testis, we detected enlarged Dpp-positive punctae in hub cells, likely reflecting the lysosomes (Fig 4K and 4L).

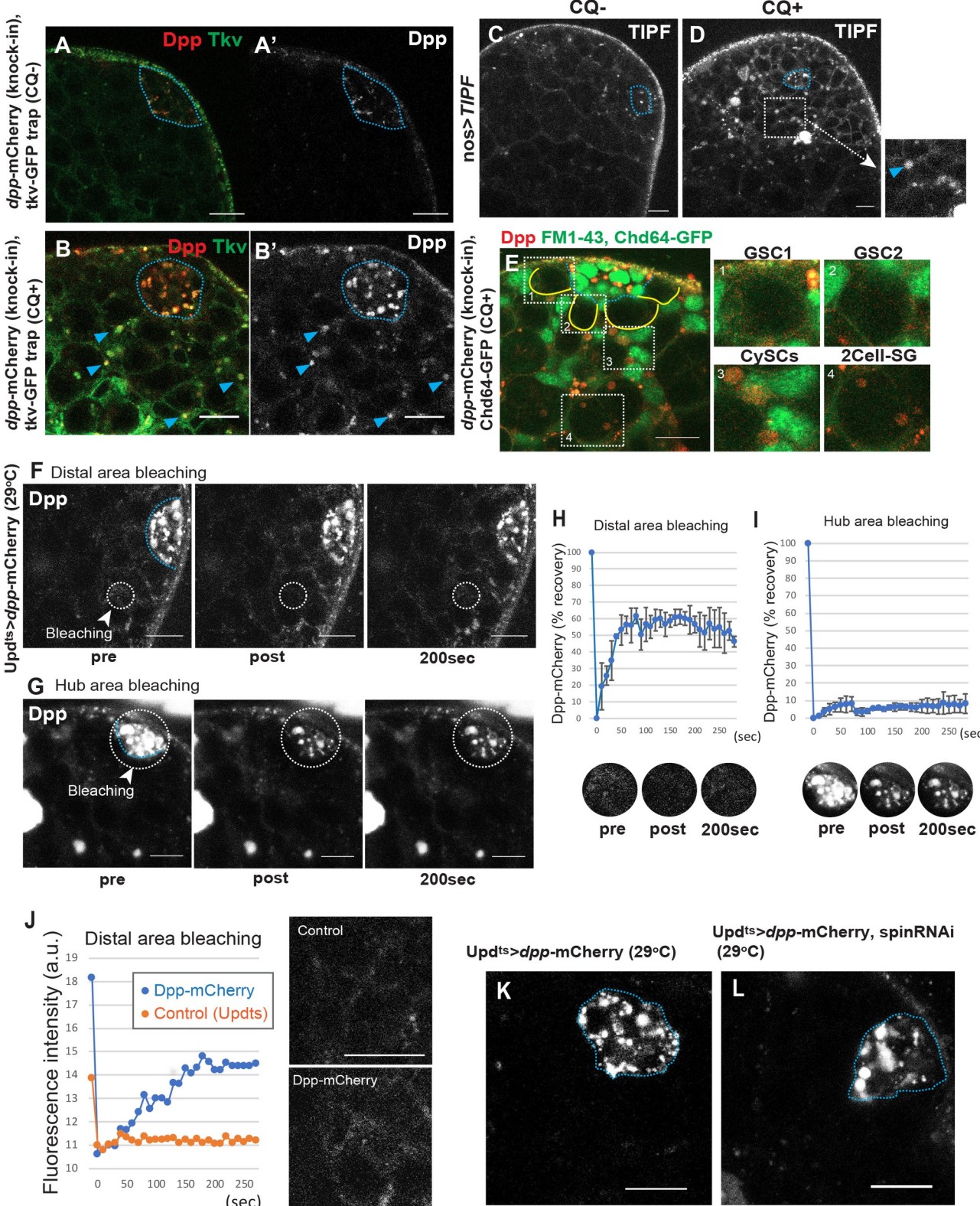

**Fig 4. Inhibition of hub lysosomes does not impact Dpp diffusion. A, B,** Representative images of testis tips of Dpp-mCherry knock-in with Tkv-GFPtrap (**A** and **B**), after 4-hour incubation without (**A**) or with (**B**) CQ. Blue dotted lines outline the hub. Blue arrowheads indicate lysosomal localization of Dpp/Tkv outside of the hub. **C, D,** Representative images of testis tips with expression of ligand-bound Tkv sensor, TIPF, under the

control of a germline-specific driver (*nosGal4>TIPF*) after 4-hour culture without (**C, CQ−**) or with (**D, CQ+**) CQ. The right panel shows a magnified image of the white square region in **D**. The blue arrowhead indicates lysosomal localization of TIPF. **E,** A representative image of the testis tip of a *dpp-mCherry* knock-in line after 4-hour incubation with CQ. Blue dotted lines outline the hub. Yellow lines outline GSCs. Red; Dpp-mCherry, Green; Chd64-GFP (CB03690, a CySC marker) and FM1-43, membrane dye. Right panels (1–4) show high magnification images of the indicated cell types. **F, G,** Representative FRAP experiments of testis tips of Dpp-mCherry expressed in the hub (*updGal4^{ts} >dpp-mCherry*) after a 4-day temperature shift. Blue dotted lines outline the hub. The regions encircled by white dotted lines were photobleached, and the intensity of the mCherry signal was monitored before and after photobleaching at the indicated time points. **H, I,** Recovery curves of the Dpp-mCherry signal after photobleaching at a distal area (**H**) or a hub area (**I**) (See Methods). Lower panels are a magnification of the white dotted circles in **F, G**. Each graph shows means and standard deviations from 3 and 2 independent FRAP experiments, respectively. **J,** Examples of distal area recovery of a Dpp-mCherry expressing testis (*updGal4^{ts}>dpp-mCherry*, blue) or a control testis (*updGal4^{ts}* only, orange) in which Dpp-mCherry is not expressed. Imaging was performed after a 4-day temperature shift. Right panels show the fluorescence of the distal area before bleaching. **K, L,** Representative images of testis tips of flies expressing Dpp-mCherry with or without Spin RNAi in hub cells (*updGal4^{ts}> dpp-mCherry, spinRNAi*). Blue dotted lines outline the hub. Scale bars are 10 μm in all images. All experiments were performed using live tissues. Underlying numerical data for **H, I,** and **J** are provided in S1 Data. CQ, chloroquine; CySC, somatic cyst stem cell; Dpp, Decapentaplegic; FRAP, fluorescence recovery after photobleaching; GSC, germline stem cell; RNAi, RNA interference; Spin, spinster; Tkv, Thickveins.

However, we did not detect increased level of diffusing Dpp signal in the spin knockdown testis tip (Fig 4K and 4L).

These observations indicate that Dpp protein can diffuse from the hub and is taken up by cells located in the field outside of the niche for subsequent degradation. Inhibition of hub lysosomes did not show a detectable change in the level of Dpp diffusion.

Together, our data suggest that hub lysosomes suppress Dpp signaling in GSCs, most likely by degrading Tkv derived from MT-nanotubes. Hub derived JAK-STAT signals and Dpp transcription and protein levels did not exhibit changes after hub lysosome inhibition and are thus unlikely to influence the pMad level in GSCs.

## Ubiquitination mediates Tkv degradation by promoting translocation of Tkv from GSCs to hub lysosomes

If Tkv is internalized from the MT-nanotube membrane into hub cells prior to its lysosomal degradation, a hub lysosomal defect should result in an accumulation of Tkv-positive vesicles inside of the hub cells, and they will no longer be usable for signaling. Therefore, it is puzzling how the impairment of hub lysosomes increases the active Tkv fraction on MT-nanotubes. To gain insight into the mechanistic aspect of Tkv transfer, we examined the effect of ubiquitination-defective Tkv.

The ubiquitination of membrane proteins generally recruits ESCRT (endosomal sorting complexes required for transport) machinery, resulting in a wide range of membrane reorganization including endocytosis, membrane budding, multivesicular body formation, and lysosomal degradation [22]. SMAD ubiquitination regulatory factor (Smurf) is a HECT (homologous to the E6-AP carboxyl terminus) domain-containing protein with E3 ubiquitin ligase activity, and disruption of Smurf enhances the Dpp-Tkv signal in GSCs [23,24]. It has been reported that phosphorylation of the Ser238 residue of Tkv is required for Smurf-dependent ubiquitination and targeting for subsequent proteolysis [23].

We found that Tkv-S238A-GFP, in which the serine residue required for Smurf-dependent ubiquitination was mutated, exhibits striking differences from wild-type Tkv-GFP on 2 accounts. When both constructs were expressed in GSCs (*nosGal4>tkv-GFP vs. nosGal4>tkv S238A-GFP*), the amount of S238A-Tkv protein was significantly higher than that of wild-type Tkv (S6A and S6B Fig). Secondly, Tkv S238A-GFP exhibited a change in localization: It strongly localized to the entire cell cortex of GSCs including the MT-nanotube membrane, while colocalization with hub lysosomes was greatly diminished (S6C–S63 Fig), suggesting that the ubiquitination of Tkv is likely required for Tkv transfer from GSCs to hub cells. Expression of Tkv-S238A resulted in an up-regulation of pMad (S6F–S6H Fig), consistent

with the model that Tkv transfer from GSCs to hub cells is required for the attenuation of Dpp signaling.

These results demonstrate that ubiquitination of Tkv is required for its translocation from GSCs to the hub lysosomes for degradation, which is critical to attenuate Dpp signaling in GSCs, as overexpression of degradation defective Tkv, but not wild-type Tkv, within GSCs is sufficient for increasing pathway activation. It is still unclear how ubiquitinated Tkv receptor is recognized by hub lysosomes and how hub lysosomes influence the transfer of Tkv from MT-nanotube to hub cell lysosomes.

## Discussion

In a previous report, we have shown that niche cells and stem cells interact in a contact-dependent manner, with GSCs and hub cells engaging in productive signaling via MT-nanotubes, enabling highly specific cell–cell interactions and excluding non-stem cells from receiving stem cell signals [8]. Here, we demonstrate MT-nanotubes also contribute to the proteolysis of a receptor via transferring the stem cell–derived receptor to lysosomes in the niche cells. This mechanism may ensure the removal of excess receptor preventing overload of the niche signal in stem cells.

We observed that Tkv overexpression reversed the effect of MT-nanotube loss on signaling outcomes (negative to positive). Our model (see Fig 2L) complementarily accounts for both of these results by proposing that MT-nanotubes promote signaling by increasing the probability of Tkv/Dpp interaction via recruiting the receptor to a confined space (i.e., the MT-nanotube surface). In this scenario, MT-nanotubes promote signaling even when the cell has only a low expression level of endogenous Tkv. Future studies will be necessary to determine the precise concentration of Tkv and Dpp molecules in each location (MT-nanotube surface versus cell body) to further understand the mechanism by which MT-nanotubes regulate the signaling.

The mechanism of Tkv internalization and degradation in ligand producing hub cells is completely unknown. Receptor endocytosis is critical for signal transduction outcomes in broad systems including BMP signaling [25–28]. Cytonemes, another type of signaling protrusion, traffic non-membrane-bound ligands from signal sending cells to signal receiving cells that are located at a distance from each other [29,30]. Cytoneme-derived ligands are also internalized into the receptor expressing cells [30,31]. However, the opposite instance, namely receptor internalization into ligand producing cells, has never been reported. In that sense, our observation of Tkv transfer between GSCs and hub cells may utilize a unique, yet unknown mechanism.

Our data do provide a few clues about the mechanism of Tkv transfer to Hub lysosomes: (1) Hub-specific knockdown of proteins required for lysosomes resulted in increased Tkv in hub lysosomes and GSC membrane (MT-nanotubes), suggesting that Tkv is transferred from GSC to hub lysosomes for degradation; (2) The observation that Tkv overexpression do not overactivate the downstream pathway suggests that ligand-free Tkv may be internalized (see model in Fig 2L); (3) The ubiquitination-defective mutant of Tkv cannot be internalized, implying that ubiquitination of Tkv is required for internalization; (4) Tkv transferred from GSC does not cause downstream signal activation in hub cells, indicating either that the Tkv cytoplasmic tail faces inside the transported vesicles or that internalized Tkv is otherwise inactivated; and (5) MT-nanotubes are required for Tkv transfer from GSC to hub cell. MT-nanotubes, although structurally distinct from cilia, require IFT proteins for formation and thus share some regulatory mechanisms with cilia.

Cilia are ancient structures that extend from the cell surface to sense and transduce various extracellular signals. Ciliary membrane is often shed as small vesicles, or an entire portion of a cilium can be also removed [32]. Such mechanisms are called "ciliary membrane shedding," "ciliary ectosomes" [33,34], "cilia ectocytosis" [35], and "cilia decapitation" [36,37]. These

mechanisms have been shown to regulate various biological processes including regulating signaling receptors [38]. In contrast to many types of cilia, MT-nanotubes penetrate neighboring hub cells and form double membrane surfaces composed of plasma membrane from 2 adjacent cell types, the GSC and the hub cell. A similar situation is well documented in the retina where a specialized cilium on photoreceptor cells, the outer segment, undergoes a daily renewal process whereby the tip of the cilium continuously sheds and is engulfed by the juxtaposed retinal pigment epithelium (RPE) cells (reviewed in [39]). Like RPE cells, hub cells may engulf membrane from the tip of the MT-nanotubes to remove Tkv. Further studies including high-resolution live imaging as well as ultrastructure analysis of MT-nanotubes and vesicles/lysosomes in hub cells will be necessary to elucidate the molecular details of this mechanism.

How does the digestion of a stem cell–derived receptor in the niche cell benefit short-range signaling? The range of the self-renewal signal from the niche must be extremely short, as the GSC daughter gonialblasts, located only one cell layer away from the hub, must enter the differentiation program. It has been proposed that heparan sulfate proteoglycans (HSPGs) are essential for GSC maintenance via concentrating hub-derived ligands including Dpp on the cell surface of hub cells [40], suggesting that a certain amount of Dpp must be trapped on the surface of hub cells. On the other hand, a previous study demonstrated the lack of a physical barrier around germ cells up to 2-cell SG stages [41], suggesting the possibility that some fraction of ligands can diffuse away. Consistent with their report, we detected the diffusion of Dpp in the extracellular space several cell diameters away from the hub.

Furthermore, we found that the diffusing Dpp is internalized and digested in the lysosomes of cells located outside of the niche. Another report has shown that CySCs express Tkv available to absorb any free Dpp [42], consistent with our observation that the majority of cells containing Dpp-positive lysosomes were CySCs. We also observed Dpp in SG lysosomes at lower intensities. Importantly, cells outside the niche, including CySCs and SGs, lack pMad staining in their nuclei indicating that they possess a mechanism to prevent activation of the downstream pathway.

In contrast to the cell outside of the niche, GSCs, require the signal for self-renewal, must be competent for signal activation. Therefore, if the Dpp-Tkv complex is internalized into GSCs, it may activate downstream pathway from signaling endosomes. Signaling endosomes can be inherited into differentiating daughter cells during the cell division. Thus, the lack of Dpp/Tkv internalization in GSCs could be the strategy to ensure turning off the signal in the daughter cell upon exit from the niche.

It remains an open question whether lysosomal proteolysis of stem cell–derived receptors in niche cells, as demonstrated by our study, might also regulate other stem cell systems. GSCs in the *Drosophila* ovary have also been reported to project cellular protrusions into the niche cell cluster to access a reservoir of Dpp [43]. Intriguingly, microtubule-rich protrusions are required for attenuation of Dpp signaling, suggesting the possibility that female GSCs utilize a similar mechanism to degrade signaling components.

## Methods

### Verification of the functionality of fluorescently tagged proteins

We acknowledge the general problems with fluorescently tagged proteins, such as truncation of the tagged portion, mislocalization/aggregation, and changes in protein stability. In addition, Gal4/UAS-mediated expression may not accurately reflect the amount of endogenous protein. Moreover, as we noted in our previous report [8], we observe that mCherry, but not GFP itself, is trafficked from GSC to hub cells, indicating the possibility that the mCherry tag may promote the trafficking regardless of the transgene. However, we had to largely rely on

fluorescently tagged proteins and live observation to determine their localization and behavior. This was due to technical difficulties caused by low levels of endogenous protein amounts and/ or difficulties in preserving structure during fixation. Thus, we summarized the functional verification of constructs used in this study here.

The Dpp-mCherry knock-in fly and Tkv-GFPtrap fly are both homozygous viable, and we did not detect any phenotype in the testis; in addition, we observed almost complete colocalization in the hub area (S1D Fig). The complete colocalization of nosGal4-driven expression of Tkv-mCherry (C terminus tagged) with Tkv-GFPtrap (N terminus tagged) indicates that both constructs likely represent the localization of full-length Tkv protein (S1E Fig). In summary, we consider these constructs to be equivalent and to reliably report the endogenous protein's behavior. Nevertheless, future studies using alternative methods might be necessary to verify the results obtained using these constructs.

## Fly husbandry and strains

All fly stocks were raised on standard Bloomington medium at 25˚C (unless temperature control was required) and young flies (0- to 7-day-old adults) were used for all experiments. The following fly stocks were used: *hs-flp; nos-FRTstop-FRT-gal4, UAS–GFP[31]; tkv-GFP* protein trap line (CPTI-002487, inserted in the first intron of Tkv-RD, a gift from B. McCabe); *tub-GFP-Lamp1* [44] (FBrf0207605, a gift from H. Krämer); *updGal4* (FBti0002638, gift from Y. Yamashita); *nosGal4* [45] (gift from Y. Yamashita); tubGal80$^{ts}$ ([46] a gift from C.Y. Lee).

UAS-TIPF [14], UAS-Dpp-mCherry [30], UAS-Tkv-mCherry [30], UAS-Tkv-GFP [30], and Dpp-mCherry (CRISPR knock-in) [13] were gifts from T. Kornberg.

The following stocks were obtained from the Vienna Drosophila Resource Center (VDRC): *oseg2 RNAi* (VDRC GD8122); *osm6 RNAi* (VDRC GD24068); *che-13 RNAi* (VDRC GD5096); *punt-GFP* (VDRC 318264, *2XTY1-SGFP-V5-preTEV-BLRP-3XFLAG*). Other stocks were from the Bloomington Stock Center: *chd64-GFP* (FlyTrap Project CB03690) [43]; *UAS–GFP–αtubulin* (BDSC 7253); *spin RNAi* (TRiP.JF02782); *lamp1 RNAi* (Line1:TRiP.HMS01802, Line2:TRiP GLV21040), *tkv-CA* (BDSC36537). For expression of Dpp-mCherry, the *updGal4$^{ts}$* driver, a comination of *updGal4* and *tubGal80$^{ts}$*, was used to avoid lethality. Temperature shift crosses were performed by culturing flies at 18˚C to avoid lethality during development and shifted to 29˚C upon eclosion for 4 days before analysis. Control crosses for RNAi screening were designed with matching gal4 and UAS copy numbers using TRiP control stock (BDSC 35785) at 25˚C.

## Quantitative RT-PCR

Females carrying a nosGal4 driver were crossed with males of indicated RNAi lines.

Testes from 100 male progenies, aged 0 to 7 days, were collected and homogenized by pipetting in TRIzol Reagent (ThermoFisher, Asheville, North Carolina), and RNA was extracted following the manufacturer's instructions. One microgram of total RNA was reverse transcribed to cDNA using SuperScript III First-Strand Synthesis Super Mix (ThermoFisher) with Oligo (dT)20 Primer. Quantitative PCR was performed, in duplicate, using SYBR green Applied Biosystems Gene Expression Master Mix on a CFX96 Real-Time PCR Detection System (Bio-Rad, Hercules, California). A control primer for αTub84B (5′-TCAGACCTCGA AATCGTAGC-3′/5′-AGCAGTAGAGCTCCCAGCAG-3′) and experimental primers for

Spin (5′-GCGAATTTCCAACCGAAAGAG-3′/5′-CGGTTGGTAGGATTGCTTCT-3′),
Lamp1 (5′-AACCATATCCGCAACCATCC-3′/5′-CCTCCCTAGCCTCATAGGTAAA-3′)
were used. Relative quantification was performed using the comparative CT method (ABI manual). Other RNAi lines were validated previously [8].

## Generation of pUASp-Tkv S238A transgenic flies

EGFP cDNA was amplified from the *Drosophila* gateway pPGW vector (https://emb.carnegiescience.edu/drosophila-gateway-vector-collection#_Copyright,_Carnegie) using the following primers with restriction sites (underlined):

BglII GFP F 5′- ACAGATCTATGGTGAGCAAGGGCGAGGAGCTGTTCA-3′
AscI GFP R 5′-TAGGCGCGCCTTACTTGTACAGCTCGTCCATGCCGAGA-3′

Products were then digested with BglII and AscI. NotI and BglII sites (underlined) were attached to a synthesized TkvS238A fragment (gBlock; Integrated DNA Technologies, Coralville, Iowa, sequence as follows):

5′-ATGCGGCCGCACCATGGCGCCGAAATCCAGAAAGAAGAAGGCTCATGCCCGC
TCCCTAACCTGCTACTGCGATGGCAGTTGTCCGGACAATGTAAGCAATGGAACC
TGCGAGACCAGACCCGGTGGCAGTTGCTTCAGCGCAGTCCAACAGCTTTACGA
TGAGACGACCGGGATGTACGAGGAGGAGCGTACATATGGATGCATGCCTCCCGAA
GACAACGGTGGTTTTCTCATGTGCAAGGTAGCCGCTGTACCCCACCTGCATGGCAA
GAACATTGTCTGCTGCGACAAGGAGGACTTCTGCAACCGTGACCTGTACCCCACC
TACACACCCAAGCTGACCACACCAGCGCCGGATTTGCCCGTGAGCAGCGAGTCCC
TACACACGCTGGCCGTCTTTGGCTCCATCATCATCTCCCTGTCCGTGTTTATGCT
GATCGTGGCTAGCTTATGTTTCACCTACAAGCGACGCGAGAAGCTGCGCAAGCA
GCCACGTCTCATCAACTCAATGTGCAACTCACAGCTGTCGCCTTTGTCACAAC
TGGTGGAACAGAGTTCGGGCGCCGGATCGGGATTACCATTGCTGGTGCAAA
GAACCATTGCCAAGCAGATTCAGATGGTGCGACTGGTGGGCAAAGGACGATA
TGGCGAGGTCTGGCTGGCCAAATGGCGCGATGAGCGGGTGGCCGTCAAGACCTTC
TTTACGACCGAAGAGGCTTCTTGGTTCCGCGAGACTGAAATCTATCAGACAGTGC
TGATGCGACACGACAATATCTTGGGCTTCATTGCCGCCGATATCAAGGGTAA
TGGTAGCTGGACACAGATGTTGCTGATCACCGACTACCACGAGATGGGCAGCC
TACACGATTACCTCTCAATGTCGGTGATCAATCCGCAGAAGCTGCAATTGCTGGC
GTTTTCGCTGGCCTCCGGATTGGCCCACCTGCACGACGAGATTTTCGGAACCCC
TGGCAAACCAGCTATCGCTCATCGCGATATCAAGAGCAAGAACATTTTGGTCAA
GCGGAATGGGCAGTGCGCTATTGCTGACTTCGGGCTGGCAGTGAAGTACAACTC
GGAACTGGATGTCATTCACATTGCACAGAATCCACGTGTCGGCACTCGACGCT
ACATGGCTCCAGAAGTATTGAGTCAGCAGCTGGATCCCAAGCAGTTTGAAGA
GTTCAAACGGGCTGATATGTATTCAGTGGGTCTCGTTCTGTGGGAGATGACCCGTC
GCTGCTACACACCCGTATCGGGCACCAAGACGACCACCTGCGAGGACTACGCCC
TGCCCTATCACGATGTGGTGCCCTCGGATCCCACGTTCGAGGACATGCACGC
TGTTGTGTGCGTAAAGGGTTTCCGGCCGCCGATACCATCACGCTGGCAGGAGGA
TGATGTACTCGCCACCGTATCCAAGATCATGCAGGAGTGCTGGCACCCGAA
TCCCACCGTTCGGCTGACTGCCCTGCGCGTAAAGAAGACGCTGGGGCGAC
TGGAAACAGACTGTCTAATCGATGTGCCCATTAAGATTGTCAGATCTCA-3′

Synthesized fragments were annealed and digested by NotI and BglII. The resultant 2 inserts (TkvS238A and GFP) were ligated to a modified pPGW vector using NotI and AscI sites in the multiple cloning sites. Transgenic flies were generated using strain attP2 by PhiC31 integrase-mediated transgenesis (BestGene Inc, Chino Hills, California).

## Generation of nos-loxP-mCherry-loxP-gal4-VP16 transgenic flies

Step 1: Construction of loxP-mCherry-SV40-loxP was performed as follows. mCherry cDNA was amplified using primers NheI mCherry Fw (5′-acgctagctatggtgagcaagggcgaggag-3′) and XhoI mCherry Rv (5′-gactcgagttacttgtacagctcgtccat-3′) from the pmCherry-C1 Vector (Takara Bio USA Inc,

Mountain View, California), and then the product was introduced into NheI-XhoI sites of the pFRT-SV40-FRT vector (a gift from Elizabeth R. Gavis). Step 2: BamHI-loxP-NotI oligo (5′-GATCCATAACTTCGTATAGCATACATTATACGAAGTTATGC-3′, 5′-GGCCGCA TAACTTCGTATAATGTATGCTATACGAAGTTATG-3′) was inserted into the BamHI NotI site after the SV40 polyA sequence of the StepI vector. NdeI-loxP-NheI oligo (5′-CATATG CAACATGATAACTTCGTATAGCATACATTATACGAAGTTATTG-3′, 5′-CTAGCAA TAACTTCGTATAATGTATGCTATACGAAGTTATCATGTTGCATATGCATG- 3′) was inserted into the NdeI/NheI site upstream of the mCherry sequence of the StepI vector. Step 3: The NotI-BamHI flanked 3.13-Kb fragment from the pCSpnosFGVP (a gift from Elizabeth R. Gavis) containing the Nanos 5′ region-ATG (NdeI-start codon) Gal4-VP16-Nanos 3′ region was subcloned into NotI-BamHI sites of pUAST-attB. Step 4: The NdeI-flanked loxP-mCherry-SV40 polyA-loxP fragment was subcloned into the NdeI start codon of the plasmid described in Step 3. A transgene was introduced into the attP2 using PhiC31 integrase-mediated transgenesis systems by BestGene, Inc.

### Live imaging

Testes from newly eclosed flies were dissected into Schneider's *Drosophila* medium containing 10% fetal bovine serum and glutamine–penicillin–streptomycin. These testes were placed onto Gold Seal Rite-On Micro Slides' 2 etched rings with media, then covered with coverslips.

Images were taken using a Zeiss LSM800 confocal microscope with a 63× oil immersion objective (NA = 1.4) within 30 minutes. Images were processed using Image J and Adobe Photoshop software. Three-dimensional rendering was performed with Imaris software.

### Immunofluorescent staining

Immunofluorescent staining was performed as described previously [47]. Briefly, testes were dissected in phosphate-buffered saline (PBS) and fixed in 4% formaldehyde in PBS for 30 to 60 minutes. Next, testes were washed in PBST (PBS + 0.3% TritonX-100) for at least 30 minutes, followed by incubation with primary antibodies in 3% bovine serum albumin (BSA) in PBST at 4˚C overnight. Samples were washed for 60 minutes (3 times for 20 minutes each) in PBST, incubated with secondary antibodies in 3% BSA in PBST at 4˚C overnight, and then washed for 60 minutes (3 times for 20 minutes each) in PBST. Samples were then mounted using VEC-TASHIELD with 4,6-diamidino-2-phenylindole (DAPI) (Vector Lab, H-1200).

The primary antibodies used were as follows: rat anti-Vasa (1:20) and mouse anti-Bam (1:20) were obtained from the Developmental Studies Hybridoma Bank (DSHB); Rabbit anti-Smad3 (phospho S423 + S425) (1:100, Abcam, ab52903); Guinea pig anti-STAT92E (1:2000, a gift from Yukiko M. Yamashita); rabbit anti-Zfh1 (1:4000; a gift from Ruth Lehmann).

AlexaFluor-conjugated secondary antibodies were used at a dilution of 1:400.

Images were taken using a Zeiss LSM800 confocal microscope with a 63× oil immersion objective (NA = 1.4) and processed using Image J and Adobe Photoshop software.

### Quantification of Dpp mRNA

Fluorescent in situ hybridization was performed as described previously [48].

Briefly, testes were dissected in 1X PBS and then fixed in 4% formaldehyde/PBS for 45 minutes. After fixing, they were rinsed 2 times with 1X PBS, then resuspended in 70% EtOH, and left overnight at 4˚C. The next day, testes were washed briefly in wash buffer (2X SSC and 10% deionized formamide), then incubated overnight at 37˚C in the dark with 50 nM of Quasar 570 labeled Stellaris probe against dpp mRNA (LGC Biosearch Technologies, a gift from Michael Buszczak [49]) in the hybridization buffer containing 2X SSC, 10% dextran sulfate

(Sigma-Aldrich Inc., St Louis, Missouri), 1 µg/µl of yeast tRNA (Sigma-Aldrich Inc.), 2 mM vanadyl ribonucleoside complex (NEB), 0.02% RNAse-free BSA (ThermoFisher), and 10% deionized formamide. On the third day, testes were washed 2 times for 30 minutes each at 37˚C in the dark in the prewarmed wash buffer (2X SSC, 10% formamide) and then resuspended in a drop of VECTASHIELD with DAPI (Vector Lab, H-1100).

For quantification of the FISH signal, z-stacks were collected at 0.5 µm intervals using the same acquisition settings of for confocal microscopy using a Zeiss LSM800. The total number of particles in the hub was counted using Octane1.5.1 (Super-resolution Imaging and Single Molecule Tracking Software (https://github.com/jiyuuchc/Octane)).

### Clone induction

For clonal expression of Tkv-mCherry, *hs-cre*, *nos-loxP-stop-loxP-Gal4*, *UAS-tkv-mCherry*, *UAS-GFP-αtubulin* flies were heat-shocked at 37˚C for 15 minutes. Testes were dissected 24 hours after the heat shock. For the time course of Tkv-mCherry localization, *hs-flp*, *nos-FRT-stop-FRT-Gal4*, *UAS-tkv-mCherry*, *UAS-GFP* flies were heat-shocked at 37˚C for 60 minutes. Testes were dissected at indicated times (day 1, 2, 3) after the heat shock.

### Chloroquine and Lysotracker/LysoSensor treatment

Testes from newly eclosed flies were dissected into Schneider's *Drosophila* medium containing 10% fetal bovine serum and glutamine–penicillin–streptomycin. Testes were then incubated at room temperature with or without 100 µM chloroquine (Sigma) in 1 mL media for 4 hours prior to imaging. For lysosome staining, testes were incubated with 50 nM of LysoTracker Deep Red (ThermoFisher L12492) or 100 nM of LysoSensor Green DND-189 (ThermoFisher L7535) probes in 1 mL media for 10 minutes at room temperature then briefly rinsed with 1 mL of media 3 times prior to imaging.

These testes were placed onto Gold Seal Rite-On Micro Slides' 2 etched rings with media, then covered with coverslips. An inverted Zeiss LSM800 confocal microscope with a 63× oil immersion objective (NA = 1.4) was used for imaging.

### Quantification of pMad intensities

Mean intensity values in a portion of GSC nuclei were measured for anti-pMad staining. To normalize the staining conditions, 3 cyst cells were randomly chosen and their average pMad intensity determined for each testis; this value was used to calculate the ratio of relative intensities (i.e., GSC/CC) for each GSC. Mean intensity values (a.u.) of cyst cells are also provided in S2A Fig.

### FRAP analysis

Fluorescence recovery after photobleaching (FRAP) of Dpp-mCherry signal was undertaken using a Zeiss LSM800 confocal laser scanning microscope with 63×/1.4 NA oil objective. Zen software was used for programming each experiment. Encircled areas of interest were photobleached using the 561 nm laser (laser power; 100%, iterations; 15). Fluorescence recovery was monitored every 10 seconds.

Background signal taken in outside of the tissue in each time point were subtracted from the signal of bleached region.

% recovery was calculated as follows:

Let $I_t$ be the intensity at each time point (t), $I_{post}$ be the intensity at post-bleaching, and $I_{pre}$ be the intensity at pre-bleaching.

The governing equation of % recovery is: % recovery = $(I_t − I_{post} / I_{pre} − I_{post}) \times 100$

Means and standard deviations from $n$ = 3 experiments are shown in each graph (Fig 4H and 4I).

Maximum value of recovery was estimated as follows. Intensities between 2 time points (40s apart each other, $I_{t+40s}$ and $I_t$) became nonsignificant ($p \geq 0.05$) at the time 40s after bleaching, thus defined as the time point reached to plateau (t = 40s, $p$ = 0.336). Maximum value of recovery was determined by averaging values of 4 points; $I_{40s}$ $I_{50s}$ $I_{60s}$ and $I_{70s}$ (53.9 ± 3.18%).

## Statistical analysis and graphing

All data are means and standard deviations. No statistical methods were used to predetermine sample size. The experiments were not randomized. The investigators were not blinded to allocation during experiments and outcome assessment. Statistical analysis and graphing were performed using Microsoft Excel 2010 or GraphPad Prism 7 software. The $P$ values from Student $t$ test or adjusted $P$ values from Dunnett multiple comparisons test are provided.

## Supporting information

**S1 Fig. (associated with Fig 1). A, B, C,** Representative images of the hub area of indicated genotypes. Arrowheads indicate the colocalization of Punt (**A**), Dpp (**B**), and TIPF (**C**) with the hub lysosomes (marked by lysotracker in **A** and **C**, lysosensor in **B**). **D, E,** Representative images of the testis tip of indicated genotypes. **D,** The Dpp-mCherry (CRISPR knock-in) signal colocalizes with the Tkv-GFPtrap signal in the hub. **E,** Tkv-mCherry expressed in the germline colocalizes with the Tkv-GFPtrap signal in the hub. **D'''** and **E'''** show magnified hub areas. Scale bars; 1 μm in **A, B, C** and 10 μm in other images. Blue dotted lines outline the hub. All experiments in this Fig were performed using live tissues. Dpp, Decapentaplegic. (TIF)

**S2 Fig. (associated with Fig 2). A, B,** Representative images of testes tips of Tkv-GFPtrap flies without (**A**) or with (**B**) IFT-KD (*nosGal4>osm6 RNAi*). Blue dotted lines outline the hub. **C–F,** Representative Bam staining (**C, D**) and STAT92E staining (**E, F**) of testes expressing Tkv-GFP with (**C, E,** *nosGal4>tkv-GFP*) or without (**D, F,** *nosGal4>tkv-GFP, osm6 RNAi*) IFT (osm6) RNAi. **G,** Representative image of STAT92E staining of the testis expressing a constitutive active form of Tkv (*nosGal4>tkv-CA*). **H, I,** Representative images of bamGal4-mediated expression of Tkv-GFP with (**H,** *bamGal4>tkv-GFP*) or without (**I,** *bamGal4>tkv-GFP, osm6 RNAi*) IFT (osm6) RNAi. Vasa (blue), pMad (red), Tkv (green). Scale bars, 10 μm. Asterisks indicate the approximate location of the hub. For **A and B,** imaging was performed using live tissues. Fixed samples were used for **C–I**. IFT, intraflagellar transport; pMad, phosphorylated Mad; RNAi, RNA interference; Tkv, Thickveins. (TIF)

**S3 Fig. (associated with Fig 3). A,** Quantification of pMad intensity in the somatic CCs (yellow arrows in Fig 3 A–D). Average intensity of CCs in the entire testis was scored from 10 testes for each genotype. ns; nonsignificant ($p \geq 0.05$), from Dunnett multiple comparisons test. **B–D,** Representative images of STAT92E (green) staining in the testis of indicated genotypes. FasIII (red, hub marker), Vasa (blue, germline marker). White dotted lines outline the hub. Yellow lines outline GSCs. **E, F,** Representative images of STAT92E staining of a broader area of the testis from indicated genotypes. Blue dotted lines outline the hub. **G–I,** Representative images of the testis tip of indicated genotypes. Zfh-1 (green, CySC marker), FasIII (red, hub marker), Vasa (blue, germline marker). **J,** Number of Zfh-1-positive CySCs in the indicated

genotypes. Testes ($n = 11$) from 2 independent experiments were scored for each group. ns; nonsignificant (p≥0.05), from Dunnett multiple comparisons test. Fixed samples were used for all experiments. Underlying numerical data for **A** and **J** are provided in S1 Data. CC, cyst cell; CySC, somatic cyst stem cell; GSC, germline stem cell; pMad, phosphorylated Mad. (TIF)

**S4 Fig. (associated with Fig 3). A–G,** Representative images of in situ hybridization using a Stellaris FISH probe against *dpp* mRNA (red) in the testes of indicated genotypes. **B,** The Dpp RNAi (negative control) testis shows almost no detectable signal in the hub, indicating the specificity of the probe. The hub is encircled by a blue dotted line. DAPI (blue) marks nuclei. Scale bars, 10 μm. **G,** Number of particles of Dpp FISH in the hub of indicated genotypes (see Methods). Data are means and standard deviations. Testes ($n = 6$) from 2 independent experiments were scored for each group. The adjusted *P* value from Dunnett multiple comparisons test is provided. ns; nonsignificant (p≥0.05). Underlying numerical data for **G** are provided in S1 Data. Dpp, Decapentaplegic; FISH, fluorescence in situ hybridization; RNAi, RNA interference. (TIF)

**S5 Fig. (associated with Fig 4).** A representative FRAP experiment of the testis tip of a control testis (updGal4^ts line, after a 4-day temperature shift) in which Dpp-mCherry is not expressed. A region encircled by a white dotted line was photobleached, and the intensity of the mCherry signal was monitored before and after photobleaching at the indicated time points. Lower panels are magnifications of the white dotted circles. Scale bars; 10 μm. Live tissues were used for imaging. Dpp, Decapentaplegic; FRAP, fluorescence recovery after photobleaching. (TIF)

**S6 Fig. Ubiquitination mediates Tkv degradation by promoting translocation of Tkv from GSCs to hub lysosomes. A, B,** Representative images of testis tips with nosGal4-mediated expression of tkv-GFP (**A**) or tkvS238A-GFP (**B**). Blue dotted lines outline the hub. The arrow in **B** indicates a MT-nanotube decorated with TkvS238A. The right panel explains the difference between Tkv and Tkv-S238A's localization pattern. **C, D,** Representative images of testis tips with nosGal4-mediated expression of tkv-GFP (**C**) or tkvS238A-GFP (**D**) (green) with lysotracker staining (red). Blue dotted lines outline the hub. **C'''** and **D'''** show magnified images of the hub area. Lysotracker-positive lysosomes (>0.5 μm diameter) positive with Tkv are marked by blue circles. **E,** Number of Tkv-positive hub lysosomes in the indicated genotypes. Lysosomes in the entire hub region were counted as lysotracker-positive punctae >0.5 μm diameter from z-stacks collected at 0.5 μm intervals. Total testes ($n = 25$) from 2 independent experiments were scored for each group. **F, G,** Representative images of pMad staining in testes with nosGal4-mediated expression of Tkv-GFP (**F**) or TkvS238A-GFP (**G**). Blue dotted lines outline the hub. Yellow lines outline GSCs. Vasa (blue), pMad (red), Tkv (green, note that Tkv overlapping with Vasa appears as cyan). Arrows indicate CCs used as an internal control (see Methods). **H,** Quantification of pMad intensity in GSCs (relative to CCs, yellow arrows in **F** and **G**) of nosGal4-mediated Tkv- or TkvS238A-expressing testes. GSCs ($n = 25$) from 2 independent experiments were scored for each group. Scale bars are 10 μm in all images. For **E** and **H**, *P* values were calculated by Student *t* tests. For **A–E,** imaging and measurements were performed using live tissues. Fixed samples were used for **F–H**. Underlying numerical data for **E** and **H** are provided in S1 Data. CC, cyst cell; GSC, germline stem cell; MT-nanotube, microtubule-based nanotube; pMad, phosphorylated Mad; Tkv, Thickveins. (TIF)

**S1 Movie. (related to Fig 1B).** 3D rendering of a portion of the hub (hub cell cortex: green) and Tkv-mCherry punctae (red).
(MP4)

**S2 Movie. (related to Fig 4F).** A representative video of recovery of the Dpp-mCherry signal after photobleaching of a region distal to the hub (Fig 4F). Images were taken every 10 seconds for 300 seconds.
(AVI)

**S3 Movie. (related to Fig 4G).** A representative video of the Dpp-mCherry signal after photobleaching of the entire hub region. Images were taken every 10 seconds for 300 seconds.
(AVI)

**S1 Data. Individual numerical values that underlie the summary data displayed in the following Fig panels, Fig 1K, 1N, 1Q, 1R; Fig 2D, 2G and 2J; Fig 3E, 3J and 3M; Fig 4H, 4I and 4J; S3A and S3J Fig; S4G Fig; and S6E and S6H Fig.**
(XLSX)

## Acknowledgments

We thank Yukiko Yamashita, Thomas B. Kornberg, Helmut Krämer, Elizabeth R. Gavis, Saugata Roy, Cheng-Yu Lee, Michael Buszczak, Ruth Lehmann the Bloomington Drosophila Stock Center, the Developmental Studies Hybridoma Bank, and the Vienna Drosophila Resource Center for reagents; Boris M. Slepchenko, Yukiko Yamashita, Laurinda Jaffe, and Michael Buszczak for discussion; Ji Yu for the guidance of image quantification; and Christopher Bonin for manuscript editing.

## Author Contributions

**Conceptualization:** Sophia Ladyzhets, Mayu Inaba.

**Funding acquisition:** Mayu Inaba.

**Investigation:** Sophia Ladyzhets, Matthew Antel, Taylor Simao, Nathan Gasek, Ann E. Cowan, Mayu Inaba.

**Methodology:** Ann E. Cowan.

**Writing – original draft:** Sophia Ladyzhets, Mayu Inaba.

**Writing – review & editing:** Matthew Antel, Taylor Simao, Ann E. Cowan, Mayu Inaba.

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
