## [Editor Report · Decision Letter 0]

29 Mar 2020

Dear Dr Inaba, 

Thank you for submitting your manuscript entitled "Niche cell lysosomes self-restrict the signaling via receptor-ligand degradation" for consideration as a Short Reports by PLOS Biology.

Your manuscript has now been evaluated by the PLOS Biology editorial staff as well as by an academic editor with relevant expertise and I am writing to let you know that we would like to send your submission out for external peer review.

Please re-submit your manuscript within two working days, i.e. by Mar 31 2020 11:59PM.

Kind regards,

Di Jiang,

Associate Editor

PLOS Biology

---

## [Decision Letter · Decision Letter 1]

7 May 2020

Dear Dr Inaba,

Thank you very much for submitting your manuscript "Stem-cell niche self-restricts the signaling range via receptor-ligand degradation" for consideration as a Short Reports at PLOS Biology. Your manuscript has been evaluated by the PLOS Biology editors, an Academic Editor with relevant expertise, and by four independent reviewers.

In light of the reviews (below), we will not be able to accept the current version of the manuscript, but we would welcome re-submission of a revised version that takes into account the reviewers' comments. Our Academic Editor and we have categorised the concerns into four groups and provided guidance on how to address them. You will find this in an attachment to this mail. In principle, we are happy for textual revisions for the concerns that can be accomplished in that way, but there are several points that the reviewers and editors feel need additional data to be convincing. We cannot make any decision about publication until we have seen the revised manuscript and your response to the reviewers' comments. Your revised manuscript is also likely to be sent for further evaluation by the reviewers.

We expect to receive your revised manuscript within 2 months. Please note given we are in the midst of the COVID-19 pandemic, we are flexible regarding turnaround time for revision.

**IMPORTANT - SUBMITTING YOUR REVISION**

*Re-submission Checklist*

*Published Peer Review*

*PLOS Data Policy*

*Blot and Gel Data Policy*

Sincerely,

Di Jiang, PhD

Associate Editor

PLOS Biology

REVIEWS:

Reviewer #1: First, this appears to be a revised manuscript. If so, I have not seen the previous version, the previous reviews or the authors' response to those reviews. So my apologies if my review covers points that the authors have already discussed.

The manuscript presents novel data that, in the Drosophila testes, the BMP receptor Tkv can be transferred from Germline Stem Cells (GSCs) to the adjacent hub cells via an unknown mechanism that may involve MT-nanotubes extending from the GSCs to the hub cells. The authors had noted these puncta in their previous publication, but had given them a different role: "Tkv-mCherry localizes along the MT-nanotubes as puncta. Furthermore, using live observation, Tkv-mCherry puncta were observed to move along the MT-nanotubes marked with GFP-αTub (Extended Data Fig. 3d), suggesting that Tkv is transported towards the hub along the MT-nanotubes." Now they show that at least some of the Tkv puncta are in lysosomes in the hub cell, along with the BMP Dpp, and have thus been transferred from the GSCs to the hub cell and internalized. 

Intriguingly, blocking lysosomal activity specifically in the hub cells increases BMP signaling in the GSCs relative to that seen in control somatic cyst cells (Fig. 2F). The authors argue that this effect is due to increased availability of Tkv in the GSCs, and perhaps also increased Dpp due to the reduced degradation in the hub cells. In other words, the transfer of Tkv/Dpp from nanotubes into hub cells reduces the levels and range of Dpp signaling from hub cells to GSCs. This is intriguing, but also somewhat opposite to their previous publication, which argued that nanotubes provide a specialized type of contact that increases BMP signal reception in the GSCs. 

The authors do not consider other, less direct mechanisms for their effect- after all, reducing lysosome activity is going to change a lot of things about hub cells, and there are additional BMP signaling components and non-BMP signaling pathways to consider. I also had problems with the tests they do present.

1) The effects of hub cell-specific lysosome inhibition

A weakness of the manuscript is that the authors do not directly test whether hub cell-specific inhibition of lysosomes increases Tkv in the GSCs, or Dpp availability. I am concentrating especially on the hub-specific result because it is the only one that is pertinent to their hypothesis, which is not about a general role for degradation, but a specific role of degradation after transfer to hub cells. 

The only data on Tkv after hub-cell-specific inhibition is the puncta sizes in the hub cell itself (Fig. 2C), with no data on Tkv in the GSCs. And the authors only follow Dpp levels after a manipulation (CQ treatment) that reduces degradation in all of the testes cells, including the hub cells and the GSCs. Not too surprisingly, this increases Dpp levels in the GSCs, but this says little about the specific role of Tkv-Dpp degradation in the hub cells. Expression of a form of Tkv that lacks a ubiquitination site in the GSCs increases Tkv levels and increases BMP signaling in the GSCs, but again this is not surprising. 

I am also unsure how it would work. The authors provide no evidence that blocking lysosome activity via spin-RNAi or lamp1-RNAi in hub cells reduces Tkv-Dpp internalization into the hub cells; both appear to accumulate in now even larger hub cell vesicles. So if Tkv in the GSCs is still being transferred into hub cells, why has BMP signaling in the GSCs increased? If instead the mechanism is via Dpp, now the Dpp is in the hub cell, in a vesicle, bound to Tkv. Somehow its failure to be degraded leads to Dpp's re-release as an active signaling protein? Additional data and discussion is needed.

Finally, the authors base their BMP signaling result of Fig. 2F on comparisons with pMad in control somatic cyst cells. Is it possible that what they are seeing is instead a reduction in cyst cell pMad? Are the cyst cells receiving any signaling from the hub cells? 

2) The role of MT-nanotubes

The authors claim that MT-nanotubes that reach from GSC cells to the hub cells promote the internalization of Tkv-Dpp by hub cells. This is certainly a tempting idea, as nanotubes have been invoked elsewhere as mechanisms for molecular transfer between cells. I was surprised, therefore, that the authors provide so little evidence for this, as most of their experiments do not involve changing MT-nanotubes. While they have good evidence that hub cells are internalizing Tkv made by GSCs, they do no provide evidence that the MT-nanotubes play a special role in this process. 

The only experiment that might provide such evidence comes from Figs 3A and 3B, but here I am relying on my own comparison because neither the text nor the figure legend make any mention of the pertinent data. The panels compare GSC-specific Tkv-GFP overexpression without and with knockdown of osm6, which they previously showed shortens MT-nanotubes. There is, as they published previously, an increase in Tkv on the GSC surfaces, which they had ascribed previously to reduced transfer of Tkv from the cell membrane into the nanotubes. There also appears to be a reduction in Tkv-GFP puncta in the hub cell. However, the authors do not point out either of these results in the main text, and the former result only in the legend. It is difficult to know whether the latter, more important result (reduced hub cell puncta, at least in this image) is real without the kind of quantification they do elsewhere. If they did get this data, however, they would have to be sure that these were lysosomal puncta, as previously they said many of the puncta were those attached to the nanotubes. If you shorten nanotubes, presumably you reduce the Tkv attached to the nanotubes, but that would not say much about transfer from GSCs into hub cells.

3) The conflicting roles of MT-nanotubes in signaling

In '15 the authors said that IFT-KD, and thus shortening of the MT-nanotubes, reduced pMad in the GSCs (Inaba et al. Fig. 4), saying this happened because nanotubes were used to carry BMP signaling. However, in this manuscript the authors say that IFT-KD, when coupled with Tkv-GFP overexpression, increased pMad compared with Tkv-GFP overexpression on its own (Fig. 3C,D), presumably because of increased Tkv levels in the GSCs. The authors never discuss the reason for this profound difference in signaling outcomes, and they need to. Both manipulations increase Tkv in the GSC cell bodies. Is it that loss of Tkv trafficking to the hub cells has completely opposite effects depending on Tkv levels (endogenous vs UAS-tkv)? This needs an explanation.

4) Mechanisms of transfer

The authors do not explain or much investigate how Tkv is moving from the GSC cell membrane to the inside the hub cells, and this weakens the manuscript. They do show that a mutant Tkv that cannot be ubiquitinated has reduced transfer to hub cell puncta, so ubiquitination may be part of the transfer process. But this does not get mentioned in the Discussion. The Discussion instead quotes some evidence that similar translocation can occur with other types of receptors in other situations. The authors say that cytonemes can transfer receptors to the signaling cells, but the reference quoted is a long review, and I had trouble finding the data. Could the authors instead quote the primary references? It is also noteworthy that here the BMP ligand is not cell-bound, which makes the mechanism of receptor transfer even more puzzling. It might also be worth noting that nanotubes have been invoked in Trogocytosis. 

It might be helpful to note somewhere what part of the Tkv protein was tagged in their three different Tkv constructs, as the reader may wonder whether the transfer is of all of Tkv (ECD and ICD), or only fragments. While the protein trap is tagged in the ECD (and is not entirely normal- look at homozygous wings), as far as I can figure out the Kornberg constructs are both tagged at the C terminus and thus the ICD. 

Reviewer #2: Building off of their previous work in identifying the importance of microtubule-based nanotubes to ensure close-range BMP signaling within the testis niche, the authors provide compelling evidence for hub cell-dependent lysosomal degradation of BMP ligand-receptor complexes in ensuring restriction of this niche signaling exclusively to the germline stem cell (GSC) population. The experiments presented throughout the manuscript are well-designed and the data (particularly photobleaching of ectopically diffusing dpp) is quite convincing. Overall, the work is a logical and thoughtful continuation of the original nanotube manuscript. The only difficulty with this work is in the author's interpretation of some of the data; specifically a lack of discussion regarding some surprising results.

Major issues:

While some experiments (particularly labeling somatic cell membranes in some genetic backgrounds) would be helpful in addressing these concerns the vast majority could likely be addressed by expanding upon the discussion. 

A major deficit in the interpretation of results in this manuscript is a lack of discussion about somatic cyst stem cells and cyst cells and the potential role they may be playing. It is well established that CySCs also produce dpp (and gbb) ligands and thus could easily also be signaling to GSCs. Particularly given the lack of staining for somatic cells throughout the results it is difficult to know how seriously the role for somatic cells should be considered.

Discussion of results regarding the link between hub-based lysosomal degradation of GSC-derived Tkv and an increase in pMAD staining/BMP signaling within GSCs. The data is all compelling but some confusing disparities need to be addressed. It is unclear how Tkv internalized in hub cells and associated with (nonfunctional) lysosomes would still be capable of transducing dpp signals to increase pMad in the GSCs. Alternative hypotheses (such as Tkv on the cortex signaling extensively and no longer being trafficked to the hub cells) should be

discussed. Additionally, if Tkv is incapable of being degraded this should not prevent its localization to MT nanotubes—yet the authors find a signficant decrease in Tkv localized to hub cells (Which at this resolution is presumably what you would expect to see). Clarification on this point would be helpful and could, potentially, compellingly expand upon the author's model. 

Related to Figure 4: It is difficult to know from the images shown that the increase in Dpp is on the germ cell membrane rather than also in somatic cells. As increased BMP signaling in the CySCs is known to cause defects (increased proliferation, competing with GSCs for niche 

occupancy; Lu et al, 2019) it would be helpful to know if BMP is indeed also increased within the somatic stem cells and their daughters. In addition, it is known that Jak/STAT signaling from the niche increases dpp expression in CySCs. It would be helpful for the authors to show that their manipulations of lysosome function within the hub is not also disrupting Jak/STAT and causing complicated downstream effects that need to be taken into account in addition to direct changes to BMP signaling occurring through loss of lysosomes in the hub.

Figure 4B arrowheads—it's not possible to know that these are germ cells without some type of counterstaining as they could just as easily be somatic cells. This is a small point speaking to a larger "issue" of somatic involvement in any of these events being completely ignored. This becomes more significant when thinking about mechanism—in all images of lysotracker (and similar), the germ cells appear to have notably fewer lysosomes than the hub. Is this true? And if so, does this contribute to the substantial increase in dpp signaling distant from the hub (as germ cells would not be able to degrade active Tkv/dpp complexes on their own)? If the dpp puncta associated with lysosomes distant from the hub in figure 4 are, in fact, within somatic cells rather than germ cells, it would bolster this particular interpretation. The reverse (that the puncta are in fact within germ cells) gives further credence to the author's preferred interpretation. Either way, having this information would make the conclusions more convincing.

Minor issues:

In Figure 1 (and for many other figures) the different cell types are not adequately indicated. Particularly for those reading this paper who do not work in this field, I think it would be quite challenging to identify where hub cells versus GSCs are present in these images. Outlines similar to those utilized in other panels of figures could be helpful, though counterstaining with a hub cell marker would be ideal.

Tkv-mCherry localizes significantly to GSC cell membranes in addition to its accumulation within hub cells (Fig1 O,N). Is this receptor on the cortex non-functional? In addition, the image of a testis 3days post-heat shock appears to have less Tkv localized to GSC membranes. Is this just 

in this particular image or is a movement from cortex to nanotubes a feature of Tkv signaling in GSCs?

Reviewer #3: Comments to Authors:

In this manuscript, the authors used a series of genetics and cell biology tools in fly testes to test a model that Tkv, the receptor of the BMP signaling pathway, is generated by germ stem cells, then transport to niche cells, where excess Tkv gets degraded by lysosome. Later in the manuscript, the authors propose that the ligand, Dpp, is also regulated similarly. The advantage for the niche cell, instead of germ cell, is to degrade the Dpp-Tkv compound, in order to attenuate the excessive BMP signaling. As the authors pointed out in the Discussion, similar context-dependent regulation of signaling pathways were reported in other systems, therefore the novelty could be the finding in this system, i.e. communication between Drosophila germ cells and hub cells.

There are a few major questions that should be considered:

-To study degradation pathways of the endogenous protein, experiments using antibodies that recognize the corresponding endogenous protein would be a lot more reliable than using tagged version, this is because most of the fluorescent protein tag is quite stable and could result in accumulation of proteins. I have some suggestions here: (1) It would be the best if experiments can be done using antibody that recognizes the endogenous protein. (2) If such a reagent is impossible, I feel these experiments should be done using less stable FP tag such as unstable GFP variants. (3) Experiments could be done and the results are compared under the condition that the Tkv level is compromised. For example, the GFP tagged Tkv could be studied when one copy of endogenous tkv is removed using heterozygotes, and RT-PCR or immnunoblot could be performed to compared the overall levels. (4) A control immnunoblot should reveal whether detected GFP or mCherry is indeed for the fusion protein (Tkv-GFP and Dpp-mCherry). (5) Overexpression of GFP or mCherry by itself could be another control. I feel some of the above suggested experiments should be done to strengthen the point of Tkv/Dpp being degraded by lysosome in hub cells to restrict only stem cells receive this signaling, which is major conclusion for this manuscript.

- The paper used pMad as a readout of the BMP signaling, which should be enriched in the stem cells, but the immunostaining in the control samples showed very minimal enrichment, if any, in the controls, such as in Figure 2G and 3E. To determine the differentiation state of the germ cells, if pMad staining has caveats, why not using other markers, such as Stat and markers for spectrosome/fusome, etc.?

- In Figure 2J, since the Tkv-S238A-GFP is driven by germ cell driver, it is not surprising to see changed localization in the germline stem cells. Here, the overexpression is a concern and such an experiment should be performed by making this mutation at the endogenous gene.

- The paper does not address the origin and regulation of other BMP signal such as Gbb. Both Gbb and Dpp can activate BMP signaling. This is a minor point and would be good to know.

- On page 5, this sentence in "neither downregulating hub lysosome (spin RNAi) nor Tkv trafficking to the hub (IFT-KD), both of which are expected to compromise Dpp-Tkv degradation in the hub, were not enough to impact differentiation." is confusing, by making a double negative, do the authors mean that these conditions were enough to impact differentiation? Then, the next sentence "However, we found that compromising Tkv trafficking to the hub (IFT-KD) combined with overexpression of Tkv (TkvOE) (referred to as IFT-KD/Tkv-OE) showed prolonged pMad staining in SG populations". However, these two combined conditions: IFT-KD + Tkv-OE is different from the previous two conditions: spin RNAi or IFT-KD, I do not quite get the points for this paragraph of comparing different conditions.

Reviewer #4: "Niche cell lysosomes self-restrict the signaling via receptor-ligand degradation" submitted to PLoS Biology by Ladyzhets S et al.

There is increasing amount of evidence showing that niche-derived molecules function over a short range on resident stem cells to promote their self-renewal, while differentiating daughters are shielded from such stemness-promoting signals. Inaba M demonstrated previously that in Drosophila testis, GSCs generate a novel type of microtubule-based nanotubes, which allows them to sense niche-derived stemness promoting ligand Dpp by transporting its receptor Tkv along these nanotubes and into the hub cell cluster. In this manuscript, the authors find that Tkv, in addition to its association with nanotubes, also localize within hub cells and colocalizes with lysosomal markers. They go on to show that these Tkv puncta are derived from GSCs and sensitive to the disruption of lysosomal function in hub cells or to lysosomal drug treatment. Interestingly, another Dpp receptor Put and hub-derived Dpp also co-localize with these lysosomal makers and signaling activation reporter TIFP is also activated in these lysosomes. Compromising lysosomal activity in hub cells results in enlarged Tkv puncta in hub cells and leads to detectable Dpp signaling activation in GBs, which is not normally detected in wt testis. They further show that Tkv.S238A, a Tkv variant resistant to Smurf-mediated ubiquitination, is accumulated on GSC membrane and less detected in hub cells (on nanotubes). Lastly, using testis in vitro culture and lysosomal inhibitor treatment, the authors show that Dpp is detected outside the hub cells. Based on these data, the authors proposed that these GSC-derived nanotubes, in addition to allow GSCs to sense niche-derived Dpp, also serve another purpose - to limit the signaling range of Dpp by means of degradation Dpp/Tkv complex. 

Overall, this is an interesting manuscript which addresses the mechanism of how the niche activity is spatially regulated in the Drosophila male GSC niche and is of general interest for the readership of PLoS Biology. Following experiments, if confirmed, should be able to improve/strengthen the conclusion present in this manuscript.

Main concerns:

The authors propose the following model: GSC-produced Tkv is transported along the MT-nanotubes into the hub region at the interface between GSC and hub cells; Tkv binds hub cell-produced Dpp and initiates downstream signaling in GSCs; the Tkv/Dpp complex is subsequently translocated from GSC surface into hub cells and degraded in hub cells via lysosome-dependent activity to limit Dpp signaling range. The data present here show that GSC-derived Tkv is transported to the hub region (Fig. 1) but no further evidence is provided to judge its cellular localization. It is not clear whether these Tkv puncta are present at extracellular space between hub cells or they are (endocytosed to) inside the hub cells. This is one of the key claims in this manuscript. To sustain their model, the authors should provide unambiguous evidence to show that Tkv puncta are indeed present inside hub cells and colocalized with hub cell-derived lysosomes. One potential approach is to conduct TEM analysis for their subcellular localization. Furthermore, if these Tkv puncta are indeed inside the hub cells via an endocytosis-mediated process, the intermediate Clathrin-bound endocytosed vesicle could also be detected within the hub cells (also see next point). If confirmed, these results will greatly support the proposed model. 

According to the proposed model, lysosomal activity in hub cells is essential for the degradation of the Tkv/Dpp complex and thus limits Dpp functional range. Some key supporting data include 1) enhanced Dpp signaling in germ cells when hub cell lysosomal activity is compromised (Fig. 2); 2) no alteration in dpp transcription under this condition (Fig. S2); and 3) Dpp can be detected outside the hub cells under in vitro culture condition in the present of lysosomal inhibitor (Fig. 4). Several concerns need to be clarified. Firstly, if the Tkv/Dpp complex is located within the hub cells as proposed by the authors, inhibition of lysosomal activity in hub cells would lead to defects in the degradation of the Tkv/Dpp complex inside the hub cells. How could it lead to expanded Dpp activity range in germ cells? Does it mean the Dpp/Tkv complex undergoes exocytosis to be secreted out of hub cells, Dpp is dissociated/released from Tkv and diffuses outside the hub to functions again? It is possible that disruption of lysosomal activity may affect other hub-derived signals. Have the authors checked the expression and activity of other signaling pathways such as the JAK/STAT signaling activity which could activate Dpp signaling upon ectopic expression. Secondly, dpp expression is of importance to the model. Although Dpp in situ data provides a qualitative measurement, quantitative measure is warranted. qPCR should be conducted to confirm dpp transcription level. It was shown in their 2015 Nature paper, disruption of MT-nanotube by IFT-KO reduces pMad signaling in GSCs without noticeable upregulation in GBs, suggesting diffusion of Dpp outside hub cells may not necessary induce signaling activation. Thirdly, FRAP recovery rate in Fig. 4 seems to be extremely fast with a full recovery around one minute. In L3 imaginal disc where Dpp is expressed at a much higher level along the anterior/posterior boundary compared to that in the hub cells, the FRAP recovery rate for a column of cells of 10um width (the diameter used in this study) right next to this boundary is more than 10 min (Kicheva A et al., Science 2007, Zhou S et al., CB 2012). It is possible in vitro CQ-treatment resulting in some background of this set of experiments. Indeed, the non-specific background is increased in these testes. A control experiment should be conducted to exclude this possibility. 

As discussed in their early paper (Inaba M et al., Nature 2015), the microtubule-nanotubes are very sensitive to fixation. It is surprised to note that the fixing method used in this manuscript does not aim to stabilize microtubules for nanotubes detection and Tkv co-localization (see Methods). It is possible that even under their previously described fixation condition with addition of a low concentration of microtubule-stabilizing drug Taxol, not all MT-nanotubes are preserved and detected at a confocal resolution due to its dynamic nature. The fixation method used in this manuscript likely disrupts MT-nanotubes and results in MT remnants. The authors should provide evidence to demonstrate that these Tkv puncta do not associate with remnants of disrupted MT-nanotubes during fixing process. 

Although some contact-dependent signaling events involving exchanges of molecules between ligand-sending and receptor-producing cells have been documented, including cytonemes in fly and Trogocytosis in vertebrate systems, these ligands (in the case of receptor endocytosed into ligand-sending cells) are membrane-bound proteins. Since Dpp is produced in the hub cells and believed to be secreted into extracellular matrix, how can extracellular Dpp bind Tkv to activate downstream signaling in GSCs then "extract" transmembrane receptor Tkv from GSCs and transfer it into hub cells for degradation? If proved, this is an intriguing piece of data. Since signaling sensor TIPF is activated in these lysosomal compartments, why downstream signaling (pMad or Dad-lacZ) is not detected in these hub cells under this circumstance? Note that the authors refer this reporter as Dpp signaling activation in other contexts including those CQ-treated samples (Fig. 4).

The novel function of MT-nanotubes described here is similar to a recent study showing that Drosophila female GSCs deploy an Actin- and MT-dependent "cytosensor" to sense and limit niche-derived Dpp molecule (Wilcockson SG and Ashe HL, Dev Cell 2019). This paper should be cited and discussed in this context.

Page 6, line 2, "…, how can IFT knockdown cause a tumor located outside of the niche?". This is an inaccurate statement. The authors show in page 5, 3rd paragraph that "… nor Tkv trafficking to the hub (IFT-KD), both of which …, were not enough to impact differentiation." The condition mentioned here should be "IFT-KD/Tkv-OE". 

Page 5, paragraph 4, the statement "… tumor formation is likely caused by a defect within GSC…" is not well supported by data presented herein. The germline tumor is observed in IFT-KO/Tkv-OE driven by nos-Gal4 but not bam-Gal4. Firstly, it's well known that nos-Gal4 is expressed highly in germline than bam-Gal4. Secondly and importantly, nos-Gal4-mediated IFT knockdown disrupts MT-nanotubes, thus potential releasing Dpp from the hub cells, while bam-Gal4-mediated IFT knockdown does not disrupt the distribution of hub cell-derived Dpp. Thus this data need to be interpreted cautiously. The elevated Dpp signaling activation in IFT-KO/Tkv-OE could be a combinatory effect of affecting microtubules (by IFT-KO) and spontaneous Dpp signaling activation (by Tkv-OE), not necessary a result of Dpp diffusion. Can the authors distinguish this from their model?

---

## [Decision Letter · Decision Letter 2]

2 Sep 2020

Dear Dr Inaba,

Thank you very much for submitting a revised version of your manuscript "Stem-cell niche limits its signal via degradation of stem-cell derived receptor" for consideration as a Short Report at PLOS Biology. Thank you also for your patience as we completed our editorial process, and please accept my apologies for the delay in providing you with our decision. This revised version of your manuscript has been evaluated by the PLOS Biology editors, the Academic Editor and two of the original reviewers.

The reviews are attached below. You will see that the reviewers find the manuscript very much improved and Reviewer 2 is now satisfied. However, Reviewer 1 has raised several points that remain to be addressed. After discussing the reviews with the academic editor, we encourage you to address all the points. While both points 2 and 4 can be addressed textually, we would like you to perform the experiments suggested in points 1 and 3.

In light of the reviews, we are pleased to offer you the opportunity to address the remaining points from the reviewers in a revised version that we anticipate should not take you very long. We will then assess your revised manuscript and your response to the reviewers' comments and we may consult the reviewers again.

We expect to receive your revised manuscript within 1 month.

**IMPORTANT - SUBMITTING YOUR REVISION**

*Resubmission Checklist*

*Published Peer Review*

*PLOS Data Policy*

*Blot and Gel Data Policy*

Sincerely,

Ines

--

Ines Alvarez-Garcia, PhD,

Senior Editor,

ialvarez-garcia@plos.org,

PLOS Biology

Reviewers’ comments

Rev. 1:

The manuscript has been extensively reworked, with additional data and some changed interpretations. Most notably, the authors now avoid discussing any role for changes in the release of Dpp as an explanation for the increased BMP signaling seen after inhibition of either lysosomal function in the hub cells, or reduction of MT-nanotubes and increased Tkv in the GSCs. Instead, the authors suggest that these effects are caused by increased levels of Tkv in the GSCs caused by reduced transfer to the hub cells.

1) Threads- The authors now provide evidence that loss of lysosomal function, either generally through CQ treatment, or in the hub through RNAi of spin or lamp1, causes an increase not only in Tkv-GFP-trap in round lysosomal puncta in the hub, but also in fainter threads. They interpret these as "likely" being Tkv along MT-nanotubes, and further suggest that much of the Tkv is on the MT-nanotube (GSC) membrane, where it would allow the GSC to receive more Dpp signaling and thus increase pMad. 

However, the authors do not co-localize threads with MT-nanotubes, saying there are technical difficulties, and this greatly weakens their argument. I can understand the difficulties in the case of upd-gal4-driven RNAi, since they cannot combine this with the nos-gal4 they usually use to drive MT-nanotube markers. However, there is no reason they could not use nos-gal4-driven MT-nanotube markers along with tkv-GFP-trap, and then treat with CQ to accentuate the threads. In ref 6 they also visualize nanotubes with antiserum against Klp10A, something they could couple with upd-driven RNAi. 

2) Mechanism?- The authors still do no explain why inhibition of Tkv degradation inside hub cells would lead to higher Tkv on the surface of the GSCs. In order for this mechanism to work, blocking lysosomal activity has to somehow reduce the transfer of Tkv from GSCs into hub cells. Once the Tkv is transferred into a hub cell it is no longer functional in the GSC, and whether or not it is degraded makes no difference to the GSC. If transfer follows the model in Fig. 5K, why would having Tkv in non-functional lysosomes reduce the hub cells' endocytosis of Tkv-containing exosomes? Or does blocking lysosomal function generally reduce endocytosis? The authors have to supply a plausible mechanism, even if it is hypothetical.

3) Indirect mechanisms- These difficulties led the reviewers to suggest less direct mechanisms for the lysosomal effects on BMP signaling in the GSCs, for instance by changing BMP (Dpp and Gbb) production in hub cells or cyst cells, or via some less direct effect through the hub cells' production of JAK ligands. The authors have added some evidence, but it could be more complete, and leaves some open questions.

They show that that Dpp mRNA levels are similar in affected hub cells, but do not say much about Dpp release or diffusion. They were unable to detect any Dpp mRNA in normal or experimental cyst cells, despite published evidence that they do make it, making the negative data difficult to assess. They did not examine Gbb in hub or cyst cells; aside from in situs, there are some Gbb-GFP lines at Bloomington that might work in the testes. The authors do show that early cyst cell development seem unaffected by hub cell-specific reduction of lysosomal activity, as visualized with Zfh-1.

However, while the authors do not see any effect in nuclear STAT, they do see a profound effect on cytoplasmic STAT staining in germ line cells quite distant from the hub (S2F vs G). The authors suggest that the increased BMP signaling has inhibited differentiation of GSC daughter cells and that this maintains high cytoplasmic STAT. But the authors need to provide some evidence for failed differentiation (Bam, as they did after IFT-KD?). And is failed differentiation known to maintain high STAT levels?

4) IFT-KO and Tkv- The authors have added, as I requested, data showing that the IFT-KD that reduces MT-nanotube length also reduces lysosomal UAS-Tkv in the hub. To answer the other reviewers' concerns about artifacts due to Tkv overexpression, it would have been helpful here to look at effects using the Tkv-GFP-trap instead of UAS-Tkv, as the former is presumably expressed at more endogenous levels. I'd also be curious to see whether or not the authors observed increased Tkv-GFP-trap on the surface of the GSCs after IFT-KD, as this would provide some evidence for or against their new model for the BMP signaling effects in a situation with endogenous Tkv levels.

I have one last concern with this section, and my apologies for not asking this in my previous review. The assumption is that IFT-KD is affecting Tkv in GSCs only through its effects on MT-nanotube length, rather than some other mechanism. However, in both ref 6, and Figs 3H and K, UAS-Tkv is strongly increased after IFT-KD in cells quite distant from the hub, and presumably out of range of MT-nanotube-mediated transfer to hub cells. At the best, this suggests a very long-lasting effect on Tkv as cells move away from the hub. At the worst, this suggests there is a completely different mechanism for Tkv stabilization that does not rely on MT-nanotubes. Is there a way to drive IFT-KD in cells only after they lose hub contact, and make sure that Tkv is not affected? If not, then I think some discussion is in order. 

Typos:

The Introduction discusses ref 6, but the ref number does not appear until the Results.

"we dpp mRNA"

"pMad leves"

Fig. S1 "Dpp-mChery"

Rev. 2:

The authors did an incredibly thorough job handling every reviewer concern. In particular, I appreciated the live imaging data that was added to clarify the location/cell type of internalized dpp.

---

## [Editor Report · Decision Letter 3]

22 Oct 2020

Dear Dr Inaba,

Thank you for submitting your revised Short Report entitled "Stem-cell niche limits its signal via degradation of stem-cell derived receptor" for publication in PLOS Biology. I have now obtained advice from the Academic Editor and consulted with the rest of the editorial team. 

We're delighted to let you know that we're now editorially satisfied with your manuscript. However, we have only realised now that you added an extra figure (Fig. 5) in the previous round of revision. The format of our Short Reports only allows four main figures, so please make one of them supplementary. Apologies for not noticing this earlier.

In addition, we would like you to consider an alternative title to improve clarity. We have come up with two posibilities:

1) "Localized stem-cell niche signalling is enabled by degradation of a stem-cell receptor"

2) "Self-limiting stem-cell niche signalling through degradation of a stem-cell receptor"

Before we can formally accept your paper and consider it "in press", we also need to ensure that your article conforms to our guidelines. A member of our team will be in touch shortly with a set of requests. As we can't proceed until these requirements are met, your swift response will help prevent delays to publication. Please also make sure to address the data and other policy-related requests noted at the end of this email.

- a cover letter that should detail your responses to any editorial requests, if applicable

*Copyediting*

*Published Peer Review History*

*Early Version*

Sincerely,

Ines

--

Ines Alvarez-Garcia, PhD,

Senior Editor,

ialvarez-garcia@plos.org,

PLOS Biology

Fig. 1K, N, Q, R; Fig. 2D, G, J; Fig. 3E, J, M; Fig. 4H, I, J; Fig. 5E, H; Fig. S3A, J and Fig. S4G

---

## [Editor Report · Decision Letter 4]

30 Nov 2020

Dear Dr Inaba,

On behalf of my colleagues and the Academic Editor, Mariana Federica Wolfner, I am pleased to inform you that we will be delighted to publish your Short Reports in PLOS Biology. 

PRODUCTION PROCESS

Before publication you will see the copyedited word document (within 5 business days) and a PDF proof shortly after that. The copyeditor will be in touch shortly before sending you the copyedited Word document. We will make some revisions at copyediting stage to conform to our general style, and for clarification. When you receive this version you should check and revise it very carefully, including figures, tables, references, and supporting information, because corrections at the next stage (proofs) will be strictly limited to (1) errors in author names or affiliations, (2) errors of scientific fact that would cause misunderstandings to readers, and (3) printer's (introduced) errors. Please return the copyedited file within 2 business days in order to ensure timely delivery of the PDF proof. 

If you are likely to be away when either this document or the proof is sent, please ensure we have contact information of a second person, as we will need you to respond quickly at each point. Given the disruptions resulting from the ongoing COVID-19 pandemic, there may be delays in the production process. We apologise in advance for any inconvenience caused and will do our best to minimize impact as far as possible.

EARLY VERSION

PRESS 

Kind regards,

Erin O'Loughlin

Publishing Editor, 

PLOS Biology

on behalf of

Ines Alvarez-Garcia,

Senior Editor

PLOS Biology